# Finding Correlated Equilibrium of Constrained Markov Game: A Primal-Dual Approach

**Ziyi Chen, Shaocong Ma, Yi Zhou**
Department of Electrical and Computer Engineering
University of Utah
Salt lake city, UT 84112
{u1276972,s.ma,yi.zhou}@utah.edu

## Abstract

Constrained Markov game is a fundamental problem that covers many applications, where multiple agents compete with each other under behavioral constraints. The existing literature has proved the existence of Nash equilibrium for constrained Markov games, which turns out to be PPAD-complete and cannot be computed in polynomial time. In this work, we propose a surrogate notion of correlated equilibrium (CE) for constrained Markov games that can be computed in polynomial time, and study its fundamental properties. We show that the modification structure of CE of constrained Markov games is fundamentally different from that of unconstrained Markov games. Moreover, we prove that the corresponding Lagrangian function has zero duality gap. Based on these results, we develop the first primal-dual algorithm that provably converges to CE of constrained Markov games. In particular, we prove that both the duality gap and the constraint violation of the output policy converge at the rate $\mathcal{O}(\frac{1}{\sqrt{T}})$. Moreover, when adopting the V-learning algorithm as the subroutine in the primal update, our algorithm achieves an approximate CE with $\epsilon$ duality gap with the sample complexity $\mathcal{O}(H^9 S A^2 \epsilon^{-4})$.

## 1 Introduction

Markov game [43] is a fundamental problem in game theory where multiple agents compete with each other by interacting with a dynamic environment. It has broad applications in diverse fields, including board games [44], inverse reinforcement learning [61], market pricing [34], etc.

In the existing literature, the goal of Markov game is often formulated as achieving Nash equilibrium (NE) – a product policy under which no agent can benefit via deviating from its own policy alone. Although NE has been shown to exist for general Markov games [22], computing NE turns out to be a PPAD-complete problem that cannot be solved in polynomial time [17, 32], except for some special Markov games with zero-sum structure [11, 16, 21, 63, 64] or potential structure [35, 58]. This further motivates researchers to consider other surrogate notions of equilibrium that are slightly weaker than NE but are computationally tractable, and correlated equilibrium (CE) is such a classic and popular surrogate notion [40] (see Section 2 for the formal definition) for which many convergent and efficient algorithms have been developed very recently [31, 37, 46].

While Markov game has been well studied, its formulation has limited applicability to real-life applications where the behavior of agents is usually subject to certain constraints. For example, in an uplink time division multiple access (TDMA) cognitive radio network, each user's total latency cannot exceed a threshold on average [62]. In an anti-jamming system, the jammer aims to jam legitimate signal transmitter under average power constraint [27]. To overcome this limitation, constrained Markov game has been introduced where agents compete with each other under local behavioral constraints encoded by state value functions [4]. However, most existing works on

36th Conference on Neural Information Processing Systems (NeurIPS 2022).

constrained Markov game only studied Nash equilibrium (NE) that is PPAD-complete to compute [3–6, 20, 24, 27–30, 45, 48, 51, 52, 55, 59, 60, 62], and a tractable notion of correlated equilibrium (CE) has not been formally defined for constrained Markov games. To the best of our knowledge, [26] is the only work that explored CE in constrained Markov games. However, they define CE based on the unconstrained Lagrangian function associated with the constrained Markov game, which does not correspond to an equilibrium of the original constrained Markov game. Therefore, **the goal of this work** is to formally define a tractable notion of CE for constrained Markov games, study its fundamental properties and develop provably convergent and computation-efficient algorithms for achieving such an equilibrium, as we further elaborate below.

To achieve this goal, we need to explore and address several fundamental problems. First, while one can generalize the definition of CE of Markov game to the constrained case, it may possess fundamentally different structures due to the presence of constraints. Specifically, as we introduce later in Definition 2.1, *modification* (of actions) is a key structure of CE. In standard Markov games (without constraints), CE is equivalent under both stochastic and deterministic modifications. However, it is unclear if such equivalence still holds in constrained Markov games, which critically affects the formal definition of CE as well as the algorithm design. Second, a classic and powerful tool to handle constrained problems is the strong duality of the associated Lagrangian, which has been established for NE of constrained Markov games [30, 62]. Thus, we are inspired to explore if strong duality holds for CE of constrained Markov games, which is a challenging problem due to the complex modification structure of CE and the nonconvex Lagrangian. Lastly, once we have a comprehensive understanding of the above two problems, we can hope for a provably convergent and efficient primal-dual algorithm for finding CE of constrained Markov games. In particular, it is much desired to analyze the non-asymptotic convergence rate and sample complexity of the proposed algorithm, as the existing algorithms for constrained Markov games either lack convergence guarantee [30] or only have asymptotic convergence guarantee [26, 62].

## 1.1 Our Contributions

We define a notion of CE for constrained Markov games as a generalization of the CE for Markov games. It is defined as an equilibrium at which no feasible stochastic modification of the joint policy from any agent alone can improve the associated value function (see Definition 3.2). Such a generalized CE turns out to possess a structure of *modification* that is fundamentally different from the CE of Markov games. Specifically, we prove in Theorem 1 that our proposed CE for constrained Markov games is strictly stronger with stochastic modification than with deterministic modification. As a comparison, the CE of Markov games is equivalent with both types of modifications. This fact motivates defining CE with stochastic modification for constrained Markov games, which further lays the foundation for developing the strong duality result and the primal-dual algorithm later.

We study the Lagrangian function associated with the proposed CE for constrained Markov games and prove that it has zero duality gap (See Theorem 3). Such a result is the key for developing a convergent primal-dual algorithm. Compared with the strong duality established in the existing works [2, 30, 42], the proof of our result for constrained Markov games requires new technical developments. To elaborate, the existing works yield Lagrangian functions that are defined over the entire unconstrained policy space. As a comparison, our Lagrangian function is defined over the set of stochastically modified policies, which corresponds to a subset of the policy space over which the Lagrangian function is nonconvex. To address this challenge, we transform the Lagrangian function from the policy space to the corresponding space of probability measures on episodes induced by the stochastically modified policies. We show that the transformed Lagrangian function is linear and the transformed space is convex and compact, which further implies the desired strong duality.

Then, we propose the first primal-dual algorithm that provably converges to CE of constrained Markov games. In particular, the primal update of our algorithm requires to compute an approximate CE of an unconstrained Markov game associated with the Lagrangian function, which can be efficiently solved by many existing algorithms, e.g., V-learning [31, 46] and Nash Value Iteration [37]. We prove that both the duality gap and the constraint violation of the output policy produced by our algorithm converge at the rate $\mathcal{O}(\frac{1}{\sqrt{T}})$, where $T$ denotes the number of iterations. This rate is comparable to that of the primal-dual algorithm for single-agent constrained reinforcement learning [18, 19]. Moreover, when adopting the V-learning algorithm as the subroutine in the primal update, our algorithm achieves an approximate CE with $\epsilon$ duality gap and constrained violation with the sample

complexity $\mathcal{O}(H^9 S A^2 \epsilon^{-4})$, where $H$ denotes the episode length, $S, A$ denote the cardinality of the state space and action space, respectively.

## 1.2 Other Related Work

**Markov game:** [53] introduced coarse correlated equilibrium (CCE) for Markov game that is weaker than CE. Many algorithms such as V-learning [31, 39, 46] and Nash Value Iteration [37] can achieve an approximate CCE with finite time convergence guarantee. Various other Markov game settings have also been studied. For example, [14, 25] studied mean-field Markov game with a large number of agents. In a Stackelberg game [8, 50], agents are partitioned into leaders and followers. The followers select the best response to the leaders' actions, and the leaders' goal is to achieve Stackelberg equilibrium which corresponds to the leaders' optimal strategy. Strategic game can be seen as a special case of Markov game without state and transition [41]. Extensive-form game [10, 23, 33, 65] can be seen as a special case of Markov game where any state can only be reached from a unique state at the previous step due to the tree structure of the game.

**Constrained reinforcement learning:** Single-agent constrained reinforcement learning (RL) is a single-agent RL problem with behavioral constraints. It is usually formulated as constrained Markov decision process [2], which can be seen as a special case of constrained Markov game with only one agent. Various primal-dual algorithms have been proposed for single-agent constrained RL [1, 2, 18, 19, 36, 47, 49, 56, 57]. Other types of algorithms include Lyapunov-based algorithms that use Lyapunov functions [12, 13], interior point methods that use logarithmic barrier functions [38], policy network with a special layer to encode constraints [15], and primal algorithms that update the policy via alternatively maximizing the objective function and minimizing the violated constraints [54].

## 2 Preliminaries of Markov Game

We consider $M$ agents playing a Markov game over a finite time horizon of episode length $H$. Specifically, at every time step $h = 1, ..., H$, the agents observe a global state $s_h \in \mathcal{S}$ of the environment, where $\mathcal{S}$ denotes the state space. Then, the agents take a joint action $a_h = [a_h^{(1)}, \ldots, a_h^{(M)}]$ following a joint stochastic policy $\pi_h(\cdot | s_{1:h}, a_{1:(h-1)})$, which corresponds to a distribution on the joint action space $\mathcal{A} := \prod_{m=1}^{M} \mathcal{A}^{(m)}$ and depends on the past states $s_{1:h} := \{s_t\}_{t=1}^{h}$ and past actions $a_{1:(h-1)} := \{a_t\}_{t=1}^{h-1}$. After that, the global state of the environment transfers to a new state $s_{h+1}$ following the state transition kernel $\mathcal{P}(\cdot | s_h, a_h)$, and each agent $m$ receives a local reward $r_{0,h}^{(m)}(s_h, a_h)$ from the environment. We note that here the joint policy $\pi_h$ can be a non-product (i.e., cannot be factorized into a product of independent policies associated with the agents) and non-Markov policy.

In the above Markov game, each agent $m$ collects its own rewards over the episodes. Denote $\rho$ as the initial state distribution, we can define the following value function associated with agent $m$.

$$V_0^{(m)}(\pi) := \mathbb{E}_\pi \Big[ \sum_{h=1}^{H} r_{0,h}^{(m)}(s_h, a_h) \Big| s_1 \sim \rho \Big], \tag{1}$$

which corresponds to the expected cumulative reward received by agent $m$ under joint policy $\pi := \{\pi_h\}_{h=1}^{H}$ and initial state $s$. The goal of agent $m$ is to optimize its own policy $\pi^{(m)} := \{\pi_h^{(m)}\}_{h=1}^{H}$ in order to maximize its associated value function $V_0^{(m)}(\pi)$. However, since every agent's value function is also affected by the other agents' policies and actions, the agents must compete with each other to gain more rewards until they reach a certain equilibrium. Here, we introduce two popular equilibrium notions that will be discussed throughout the paper.

**Nash Equilibrium (NE).** A joint policy $\pi$ is called an NE if the following two conditions are met: (i) for any time step $h$, the joint policy $\pi_h$ is a product policy that can be factorized into a product of *independent* policies, i.e., $\pi_h = \pi_h^{(1)} \times \ldots \times \pi_h^{(M)}$ so that the agents take independent actions; (ii) For any agent $m$ with any associated policy $\widetilde{\pi}^{(m)}$, it holds that $V_0^{(m)}(\pi) \geq V_0^{(m)}(\widetilde{\pi}^{(m)} \times \pi^{(\backslash m)})$. Here, $\pi^{(\backslash m)}$ denotes the joint policy of all the other agents excluding the agent $m$, and '$\times$' means that $\widetilde{\pi}^{(m)}$ is independent from $\pi^{(\backslash m)}$.

In other words, at NE, no agent can improve its value function by deviating from its associated policy. In the existing literature, it has been shown that computing NE is a PPAD-complete problem [17, 32],

for which it is not possible to develop polynomial-time algorithms. This observation has motivated researchers to propose a surrogate correlated equilibrium (CE) notion [40]. Before introducing the formal definition of CE, we first define the following stochastic modification operator.

**Definition 2.1** (Stochastic modification). *For any time step $h$, denote $a_h^{(m)}$ as agent $m$'s action induced by joint policy $\pi_h$. Given the past states and actions $s_{1:h}, a_{1:(h-1)}$, a stochastic modification $\phi_h^{(m)}$ (associated with agent $m$) is a distribution that randomly maps $a_h^{(m)}$ to another action $\widetilde{a}_h^{(m)}$, i.e., $\widetilde{a}_h^{(m)} \sim \phi_h^{(m)}(\cdot | s_{1:h}, a_{1:h-1}, a_h^{(m)})$. Moreover, we denote $\phi_h^{(m)} \circ \pi_h$ as the joint policy modified by $\phi_h^{(m)}$, i.e., $\pi_h$ first generates a joint action $a_h := [a_h^{(m)}, a_h^{(\backslash m)}]$, and then $\phi_h^{(m)}$ modifies $a_h^{(m)}$ to another $\widetilde{a}_h^{(m)}$ at random. We also denote $\phi^{(m)} := \{\phi_h^{(m)}\}_{h=1}^H$, $\phi^{(m)} \circ \pi := \{\phi_h^{(m)} \circ \pi_h\}_{h=1}^H$.*

**Correlated Equilibrium (CE).** A joint policy $\pi$ is called a CE if for any agent $m$ and any stochastic modification $\phi^{(m)}$, it holds that $V_0^{(m)}(\pi) \geq V_0^{(m)}(\phi^{(m)} \circ \pi)$.

Intuitively, at CE, no agent can improve its value function by modifying its own action induced by the joint CE policy. Compare to NE policies, CE policies do not require joint independence among all the agents. In fact, it has been shown that any NE policy is guaranteed to be a CE policy [31, 37, 46], and hence CE is a weaker equilibrium notion than NE. Moreover, CE can be reformulated as linear programming and therefore is computationally tractable.

# 3 Correlated Equilibrium of Constrained Markov Game

In this section, we introduce constrained Markov game – a generalization of the standard Markov game with constraints on agents' behavior. We then propose and study a new notion of correlated equilibrium for constrained Markov games.

## 3.1 Constrained Markov Game

In many real-life Markov games, the behavior of the agents is usually restricted by certain constraints, e.g., constraints on total time delay, average power, etc. This motivates us to consider a constrained version of Markov game [2]. Constrained Markov game follows the same state/action transition as that of the standard Markov game introduced in Section 2. The key difference is that, after the state transition at each time step $h$ in a constrained Markov game, every agent $m$ receives $d_m + 1$ number of rewards $\{r_{j,h}^{(m)}(s_h, a_h)\}_{j=0}^{d_m}$. Here, the accumulation of the reward $r_{0,h}^{(m)}$ corresponds to the target that agent $m$ aims to maximize. The other rewards $\{r_{j,h}^{(m)}\}_{j=1}^{d_m}$ are introduced to encode the behavioral constraints of agent $m$. In this paper, we consider the type of covering constraints on the accumulation of the rewards $\{r_{j,h}^{(m)}(s_h, a_h)\}_{j=1}^{d_m}$ over the episode, which impose constraints on the long-term behavior of agents. More specifically, define the following value functions associated with the reward functions of agent $m$:

$$V_j^{(m)}(\pi) := \mathbb{E}_\pi \Big[ \sum_{h=1}^H r_{j,h}^{(m)}(s_h, a_h) \Big| s_1 \sim \rho \Big], \quad j = 0, \ldots, d_m. \tag{2}$$

Every agent $m$ aims to optimize its own policy $\pi^{(m)}$ via the following constrained Markov game.

$$\text{(Constrained Markov Game):} \quad \max_{\pi^{(m)}} V_0^{(m)}(\pi), \ \text{ s.t. } V_j^{(m)}(\pi) \geq c_j^{(m)}, \ j = 1, \ldots, d_m. \tag{3}$$

where $\{c_j^{(m)}\}_{j=1}^{d_m} \in \mathbb{R}$ denote the thresholds of the constraints $V_j^{(m)}(\pi) \geq c_j^{(m)}$. Intuitively, the constraints on the value functions implicitly impose constraints on the joint policy $\pi$, and all the agents aim to maximize their own target value functions and achieve a certain equilibrium under their local constraints.

## 3.2 Nash Equilibrium and Correlated Equilibrium

For constrained Markov games, we can define the following notion of Nash equilibrium (NE) that is similar to the standard Nash equilibrium but takes into account the constraints [3, 27, 45, 51, 55, 60].

**Definition 3.1** (NE of constrained Markov game). *For a constrained Markov game, a joint policy $\pi$ is called an NE if the following conditions are met: (i) for any time step $h$, the joint policy $\pi_h$ is a product policy that can be factorized into a product of independent policies; (ii) $\pi$ is feasible, i.e., it satisfies the constraints of all the agents; (iii) For any agent $m$ with any associated policy $\widetilde{\pi}^{(m)}$ such that $\widetilde{\pi}^{(m)} \times \pi^{(\backslash m)}$ is feasible, it holds that $V_0^{(m)}(\pi) \geq V_0^{(m)}(\widetilde{\pi}^{(m)} \times \pi^{(\backslash m)})$.*

Compared with the NE of Markov game defined in Section 2, NE of constrained Markov game further requires the joint policies $\pi$ and $\widetilde{\pi}^{(m)} \times \pi^{(\backslash m)}$ to be feasible. It has been shown that such an NE exists for general constrained Markov games [4]. Moreover, since constrained Markov game generalizes the Markov game, computing NE of constrained Markov game is also a PPAD-complete problem. Therefore, we are motivated to propose the following surrogate notion of correlated equilibrium (CE).

**Definition 3.2** (CE of constrained Markov game). *For a constrained Markov game, a joint policy $\pi$ is called a CE if the following conditions hold: (i) $\pi$ is feasible, i.e., it satisfies the constraints of all the agents; (ii) for any agent $m$ and any stochastic modification $\phi^{(m)}$ such that $\phi^{(m)} \circ \pi$ is feasible, it holds that $V_0^{(m)}(\pi) \geq V_0^{(m)}(\phi^{(m)} \circ \pi)$.*

In particular, if $\pi$ is a product policy, it can be seen that $\phi^{(m)} \circ \pi$ is also a product policy and then CE would reduce to NE. Therefore, the above definition of CE generalizes NE of constrained Markov games. We note that [26] also defined CE for constrained Markov games, but in a very different aspect. Specifically, they only defined CE based on the **unconstrained** Markov game associated with a Lagrange function (i.e., with surrogate reward defined in eq. (8)), which does not necessarily require $\pi$ and $\phi^{(m)} \circ \pi$ to be feasible but our Definition 3.2 requires. Therefore, the existing CE proposed in [26] essentially follows the CE of **unconstrained** Markov games, whose definition is stated after our Definition 2.1. Moreover, [26] defined CE based on the state-action value function and consider stationary Markov policies in the discounted infinite horizon setting, whereas we define CE based on the state value function and consider non-Markov policies in the finite horizon setting.

It may seem that the CE in Definition 3.2 is a straightforward generalization of that of the standard Markov game. However, it turns out to possess a fundamentally different structure of the modification operator at the equilibrium point due to the presence of constraints, as elaborated in the following fundamental result. Throughout the paper, we call a stochastic modification operator deterministic if its pdf is supported on a single action.

**Theorem 1** (Modification of CE).

1. *For unconstrained Markov games. If a policy $\pi$ satisfies that $V_0^{(m)}(\pi) \geq V_0^{(m)}(\phi^{(m)} \circ \pi)$ for any agent $m$ and any deterministic modification $\phi^{(m)}$, then $\pi$ must be a CE.*

2. *For constrained Markov games. Even if a feasible policy $\pi$ satisfies that $V_0^{(m)}(\pi) \geq V_0^{(m)}(\phi^{(m)} \circ \pi)$ for any agent $m$ and any feasible deterministic modification $\phi^{(m)}$, $\pi$ may not be a CE.*

Theorem 1 shows that the structure of modification at the CE of Markov games is different from that of constrained Markov games. To elaborate item 1, recall that a CE policy $\pi$ of a Markov game is defined to satisfy $V^{(m)}(\pi) \geq V^{(m)}(\phi^{(m)} \circ \pi)$ for any stochastic modification $\phi^{(m)}$ (see Section 2). Since stochastic modifications include deterministic modifications, item 1 proves that CE of Markov games can always be achieved by a deterministic modification. Therefore, it suffices to define CE of Markov games using deterministic modifications, as adopted by many existing works [31, 37, 46]. As a comparison, item 2 shows that for constrained Markov games, the CE defined using deterministic modification is in general weaker than the CE defined using stochastic modification (see Definition 3.2), which is the focus of this paper. More importantly, as we show later in Theorem 2, defining CE of constrained Markov games using stochastic modification allows us to establish a strong duality result, which is the key to develop convergent primal-dual algorithms.

The proof of Theorem 1 is non-trivial and we provide a proof sketch here. To prove item 1, our strategy is to construct a simple deterministic modification $\phi^{(m)}$ (see Appendix C for the construction) for any optimal stochastic modification $\widetilde{\phi}^{(m)}$ and show that they achieve the same optimal value function, i.e., $V_0^{(m)}(\phi^{(m)} \circ \pi) = V_0^{(m)}(\widetilde{\phi}^{(m)} \circ \pi)$. To show this, we transform the value functions from the policy space $\pi$ to the space of probability measure $p_\pi$ of episodes induced by the associated policy, i.e., $V_0^{(m)}(\pi) = \widetilde{V}_0^{(m)}(p_\pi)$ (see eq. (22)), so that the transformed value function $\widetilde{V}_0^{(m)}$ is linear in $p_\pi$. Then we consider the extrapolated probability measure $p_\lambda := \lambda p_{\phi^{(m)} \circ \pi} + (1 - \lambda) p_{\widetilde{\phi}^{(m)} \circ \pi}$ and show that there exists $\lambda < 0$ such that $p_\lambda$ is a measure induced by a modified policy $\phi_\lambda^{(m)} \circ \pi$ for

certain proper stochastic modification $\phi^{(m)}_\lambda$. This together with the linearity of $\widetilde{V}^{(m)}_0$ proves item 1. To prove item 2, we construct a counterexample of constrained Markov game such that its CE under stochastic modification is shown to be strictly stronger than its CE under deterministic modification.

# 4 A Primal-Dual Algorithm for Constrained Markov Game

In this section, we develop a primal-dual algorithm for finding correlated equilibrium (CE) of general constrained Markov games. Throughout this section, to simplify the notation, we denote $V^{(m)}(\pi) := [V^{(m)}_1(\pi), ..., V^{(m)}_{d_m}(\pi)]$ as the collection of all the value functions in the constraints, and denote $c^{(m)} := [c^{(m)}_1, ..., c^{(m)}_{d_m}]$ as the collection of all the thresholds of the constraints.

## 4.1 Strong Duality for Correlated Equilibrium

Consider any feasible policy $\pi$ of a constrained Markov game, i.e., $V^{(m)}(\pi) \geq c^{(m)}$ holds entrywise. The CE of constrained Markov game is related to the following constrained optimization problem.

$$\text{For } m = 1, ..., M, \qquad \max_{\text{stochastic } \phi^{(m)}} V^{(m)}_0(\phi^{(m)} \circ \pi) \quad \text{s.t.} \quad V^{(m)}(\phi^{(m)} \circ \pi) \geq c^{(m)}, \qquad (4)$$

where '$\geq$' applies entrywise and the maximization is taken over all stochastic modifications. To further explain, define the following duality gap for every agent $m$.

$$D^{(m)}(\pi) := \max_{\text{feasible } \phi^{(m)}} V^{(m)}_0(\phi^{(m)} \circ \pi) - V^{(m)}_0(\pi), \qquad (5)$$

where a modification $\phi^{(m)}$ is called feasible if $\phi^{(m)} \circ \pi$ satisfies the constraint in eq. (4). Clearly, we have $D^{(m)}(\pi) \geq 0$ as the identity modification (i.e., $\phi^{(m)} \circ \pi = \pi$) is feasible. Moreover, it can be seen that $\pi$ is a CE of constrained Markov game if and only if $\max_m D^{(m)}(\pi) = 0$.

Since the problem in eq. (4) takes a constrained form, it is natural to consider its equivalent Lagrangian formulation, which is written below.

$$\max_{\phi^{(m)}} \min_{\lambda^{(m)} \in \mathbb{R}^{d_m}_+} L^{(m)}(\phi^{(m)} \circ \pi, \lambda^{(m)}) := V^{(m)}_0(\phi^{(m)} \circ \pi) + \lambda^{(m)\top}\left(V^{(m)}(\phi^{(m)} \circ \pi) - c^{(m)}\right), \quad (6)$$

where $\lambda^{(m)} \in \mathbb{R}^{d_m}_+$ is the dual variable. In order to develop convergent primal-dual algorithms for solving the constrained problem, a fundamental problem that needs to be addressed is the duality gap of the above Lagrangian function. In conventional constrained convex optimization [9], strong duality can be achieved under a Slater's condition, which requires the existence of a strictly feasible point under the constraints. Recently, this condition has also been introduced to establish strong duality for single-agent constrained reinforcement learning [18, 19, 42]. For constrained Markov game, we introduce the following Slater-type condition to study its duality gap.

**Assumption 1** (Slater's condition). *For any agent $m$ and any joint policy $\pi$, there exists a stochastic modification $\phi^{(m)}$ such that $V^{(m)}(\phi^{(m)} \circ \pi) - c^{(m)} \geq \xi^{(m)}$ for some strictly positive $\xi^{(m)}$.*

The above Slater's condition assumes that agents can always find a strictly feasible stochastic modification, regardless of the joint policy. Such a condition is closely related to the existing Slater-type condition for constrained Markov games [3, 4, 45], which replaces the modification operator $\phi^{(m)}$ by an independent local policy $\widetilde{\pi}^{(m)}$ and is used to study NE. With the above Slater's condition, we are able to establish the following strong duality result for constrained Markov game.

**Theorem 2** (Strong duality). *Let Assumption 1 hold. Then, strong duality holds for constrained Markov games, i.e., for any joint policy $\pi$ and any agent $m$,*

$$\max_{\phi^{(m)}} \min_{\lambda^{(m)} \in \mathbb{R}^{d_m}_+} L^{(m)}(\phi^{(m)} \circ \pi, \lambda^{(m)}) = \min_{\lambda^{(m)} \in \mathbb{R}^{d_m}_+} \max_{\phi^{(m)}} L^{(m)}(\phi^{(m)} \circ \pi, \lambda^{(m)}). \qquad (7)$$

To the best of our knowledge, such a strong duality result has not been established before for CE of constrained Markov games, and later we show that this is essential for developing a primal-dual algorithm for finding CE of constrained Markov games. In particular, compared to the strong duality results of single-agent constrained RL [2, 42] and those for finding Nash equilibrium (NE) of

constrained Markov game [30], the proof of Theorem 1 for finding CE of constrained Markov game is substantially different and more challenging, as we elaborate below.

**Technical novelty.** Note that the constrained settings considered in the aforementioned existing works [2, 30, 42] lead to Lagrangian functions defined over the entire unconstrained policy space, which is clearly convex and helps establish the strong duality result. As a comparison, in order to find CE, our Lagrangian function $L^{(m)}$ is a nonlinear function in $\phi^{(m)}$ and is defined over the set of modified policies $\{\phi^{(m)} \circ \pi : \phi^{(m)} \text{ is stochastic modification}\}$ for a fixed $\pi$. Such a set is a subset of the entire policy space and is not easy to analyze directly. To overcome this challenge, we first rewrite $L^{(m)}(\phi^{(m)} \circ \pi, \lambda^{(m)})$ as $\widetilde{L}^{(m)}(p_{\phi^{(m)} \circ \pi}, \lambda^{(m)})$ (see eq. (23)), which is a function of $p_{\phi^{(m)} \circ \pi}$ – the probability measure of the episode $(s_{1:H}, a_{1:H})$ induced by the modified policy $\phi^{(m)} \circ \pi$.[1] Since we showed that $\widetilde{L}^{(m)}$ is linear in both of its arguments, strong duality would follow if we can prove that the domain of the first argument $\mathcal{X} := \{p_{\phi^{(m)} \circ \pi} : \phi^{(m)} \text{ is stochastic modification}\}$ is a convex and compact set, and the proof of this result requires substantial effort. More specifically, to prove convexity we need to show that the convex combination $\lambda p_{\phi^{(m)} \circ \pi} + (1 - \lambda) p_{\widetilde{\phi}^{(m)} \circ \pi}$ for any $\lambda \in [0, 1]$ and any modifications $\phi^{(m)}, \widetilde{\phi}^{(m)}$ is a valid probability measure of the episode induced by a certain modified policy, and to prove compactness we need to show the closedness of any converging sequence $\{p_{\phi^{(m)}_{[k]} \circ \pi}\}_k$. To prove these claims, we need to study the fundamental properties of the measure $p_\pi$ and the modified measure $p_{\phi^{(m)} \circ \pi}$ in Lemma A.1 and Lemma B.1.

## 4.2 A Primal-Dual Algorithm for Finding Correlated Equilibrium

Based on the strong duality result in Theorem 2, finding CE of constrained Markov game is equivalent to solving the minimax optimization problem in eq. (7) for all agents $m$, which can be effectively solved by a primal-dual algorithm that alternatively updates the primal variable $\phi^{(m)}$ and the dual variable $\lambda^{(m)}$. While $\lambda^{(m)}$ can be updated using the standard projected dual descent update, the update of $\phi^{(m)}$ requires further exploration as we elaborate below.

Specifically, with a fixed $\lambda^{(m)}$, the primal update needs to solve the optimization problem $\max_{\phi^{(m)}} L^{(m)}(\phi^{(m)} \circ \pi, \lambda^{(m)})$. Denote $r_h^{(m)}(s, a) := [r_{1,h}^{(m)}(s, a), \ldots, r_{d_m, h}^{(m)}(s, a)]^\top$ as the collection of the rewards associated with the constraints, and define the following weighted surrogate reward $R_{\lambda, h}^{(m)}$ and its associated value function $V_\lambda^{(m)}$.

$$R_{\lambda,h}^{(m)}(s_h, a_h) := r_{0,h}^{(m)}(s_h, a_h) + \lambda^{(m)\top} r_h^{(m)}(s_h, a_h), \tag{8}$$

$$V_\lambda^{(m)}(\pi) := \mathbb{E}_\pi \Big[ \sum_{h=1}^H R_{\lambda,h}^{(m)}(s_h, a_h) \Big| s_1 \sim \rho \Big]. \tag{9}$$

Then, the optimization problem of the primal update can be rewritten as follows.

$$\max_{\phi^{(m)}} L^{(m)}(\phi^{(m)} \circ \pi, \lambda^{(m)}) = \max_{\phi^{(m)}} \Big\{ V_\lambda^{(m)}(\phi^{(m)} \circ \pi) \Big\} - \lambda^{(m)\top} c^{(m)}. \tag{10}$$

Interestingly, the problem $\max_{\phi^{(m)}} V_\lambda^{(m)}(\phi^{(m)} \circ \pi)$ can be understood as finding CE of an unconstrained Markov game with associated reward $R_{\lambda,h}^{(m)}$ and value function $V_\lambda^{(m)}$. Therefore, to perform the primal update, we can apply any black-box algorithm for finding CE of unconstrained Markov games to update the policy. In particular, some existing popular algorithms include the V-learning algorithm [31, 46] and the Nash Value Iteration algorithm [37], both of which can output a policy that achieves $\epsilon$-duality gap (defined in eq. (11)) with provable finite-time complexities.

We summarize this primal-dual algorithm in Algorithm 1, which alternatively updates the policy $\pi_t$ via finding an approximated CE of an unconstrained Markov game (with fixed $\lambda_t^{(m)}$), and updates $\lambda_t^{(m)}$ via projected dual descent (with fixed $\pi_t$). In particular, the projection set $\Lambda^{(m)} \subset \mathbb{R}_+^{d_m}$ must contain the optimal dual variable and also be bounded to ensure a bounded surrogate reward $R_{\lambda,h}^{(m)}$. We give a specific choice of $\Lambda^{(m)}$ in the next convergence analysis Section 5.

---

[1] [30, 42] use occupation measure induced by policy instead of the joint probability measure $p_\pi$, as they consider stationary Markov policies that select action $a_t$ based on the current state $s_t$.

**Algorithm 1** Primal-Dual Algorithm for Finding CE of Constrained Markov Game

---

**Input:** $\epsilon, \eta > 0; \Lambda^{(m)} \subset \mathbb{R}_+^{d_m}$ for all $m = 1, ..., M$.

**Initialize:** $\lambda_0^{(m)} = 0$ for all $m = 1, ..., M$.

**for** *iterations* $t = 0, 1, \ldots, T - 1$ **do**

- Let $\lambda^{(m)} \leftarrow \lambda_t^{(m)}$. Solve unconstrained Markov game with rewards $R_{\lambda,h}^{(m)}$ (defined in eq. (8)) and obtain an approximate CE policy $\pi_t$ that satisfies

$$\max_{\phi^{(m)}} V_\lambda^{(m)}(\phi^{(m)} \circ \pi_t) - V_\lambda^{(m)}(\pi_t) \leq \epsilon, \quad \forall m = 1, ..., M. \tag{11}$$

- Update $\lambda_t^{(m)}$ via the following projected dual descent step

$$\lambda_{t+1}^{(m)} = \text{proj}_{\Lambda^{(m)}}\left(\lambda_t^{(m)} - \eta\left(V^{(m)}(\pi_t) - c^{(m)}\right)\right), \quad \forall m = 1, ..., M. \tag{12}$$

**end**

**Output:** $\pi_{\widetilde{t}}$ where $\widetilde{t}$ is sampled from $\{0, 1, \ldots, T - 1\}$ uniformly at random.

---

**Comparison to the existing art.** Some existing works have developed other primal-dual algorithms with different primal policy improvement updates for constrained Markov games. For example, value iteration (Algorithm 11.1 of [62]), simultaneous perturbation stochastic approximation (Algorithm 11.2 of [62]) and policy gradient [30] are used as the primal update to find NE of constrained Markov games, and Q-learning [26] is used as the primal update to find CE of the Lagrangian function associated with the constrained Markov game. However, none of these works have established non-asymptotic convergence rate and sample complexity for the proposed algorithms. As a comparison, our Algorithm 1 can use any black-box primal update that can compute an approximate CE of unconstrained Markov games, and enjoys a provable non-asymptotic convergence rate and low sample complexity as we show in the next section.

## 5 Convergence Analysis and Sample Complexity

In this section, we analyze the non-asymptotic convergence rate of the primal-dual Algorithm 1. Then, we adopt the V-learning algorithm [31] as the primal update of Algorithm 1, and derive the overall sample complexity required by Algorithm 1 to achieve an approximate CE with $\epsilon$ duality gap.

We adopt the following assumption of bounded reward that is commonly adopted in the analysis of constrained RL [18, 19, 42], Markov game [31, 37, 46] and constrained Markov game [26].

**Assumption 2** (Bounded reward). *For all agents* $m = 1, ..., M$, *there exists constant* $r_{j,\max}^{(m)} > 0$ *such that* $0 \leq r_{j,h}^{(m)} \leq r_{j,\max}^{(m)}$ *for all* $1 \leq h \leq H$ *and* $0 \leq j \leq d_m$.

Under Assumptions 1 and 2, we choose the following projection set $\Lambda^{(m)}$ for Algorithm 1. In particular, we prove in Lemma E.1 that such a set contains the optimal dual variable.

$$\Lambda^{(m)} := \left[0, \frac{2Hr_{0,\max}^{(m)}}{\xi_1^{(m)}}\right] \times \left[0, \frac{2Hr_{0,\max}^{(m)}}{\xi_2^{(m)}}\right] \times \ldots \times \left[0, \frac{2Hr_{0,\max}^{(m)}}{\xi_{d_m}^{(m)}}\right] \subset \mathbb{R}_+^{d_m}. \tag{13}$$

**Theorem 3** (Convergence rate). *Let Assumptions 1 and 2 hold. Run Algorithm 1 for $T$ iterations with stepsize $\eta = \frac{1}{\sqrt{T}}$ and projection set $\Lambda^{(m)}$ defined in eq. (13). Then, the output policy $\pi_{\widetilde{t}}$ achieves the following duality gap (defined in eq. (5)).*

$$\mathbb{E}\left[D^{(m)}(\pi_{\widetilde{t}})\right] \leq \frac{2(HR_{\max}^{(m)})^2}{\sqrt{T}} + \epsilon, \quad \forall m = 1, ..., M, \tag{14}$$

*where* $R_{\max}^{(m)} := \sqrt{\sum_{j=1}^{d_m}(r_{j,\max}^{(m)})^2}$. *Moreover, define the constraint violation of the output policy $\pi_{\widetilde{t}}$ as* $W^{(m)}(\pi_{\widetilde{t}}) := \sum_{j=1}^{d_m}(\xi_j^{(m)})^{-1}\left(c_j^{(m)} - V_j^{(m)}(\pi_{\widetilde{t}})\right)_+$, *where* $(a)_+ := \max\{0, a\}$. *We obtain that*

$$\mathbb{E}\left[W^{(m)}(\pi_{\widetilde{t}})\right] \leq \frac{2HR_{\max}^{(m)}}{\sqrt{T}}\sum_{j=1}^{d_m}(\xi_j^{(m)})^{-2} + \frac{2H(R_{\max}^{(m)})^2}{r_{0,\max}^{(m)}\sqrt{T}} + \frac{\epsilon}{Hr_{0,\max}^{(m)}}, \quad \forall m = 1, ..., M. \tag{15}$$

The above theorem shows that both the duality gap and the constraint violation associated with the output policy converge at the rate $\mathcal{O}(\frac{1}{\sqrt{T}} + \epsilon)$, where $\epsilon$ is the target duality gap of the unconstrained Markov game in the primal update. In particular, a larger slackness $\{\xi_j^{(m)}\}_{j,m}$ implies a faster convergence of the constraint violation. These convergence rates are comparable to those of primal-dual algorithms for single-agent constrained RL [18, 19, 36], and the analysis is a generalization of the single-agent case. Specifically, the updated policy $\pi_t$ in the primal-dual algorithm for single-agent constrained RL is an approximate maximizer of the value function under the constraints, whereas the $\pi_t$ in our case corresponds to an approximate CE at which all agents can benefit little (up to $\epsilon$) by modifying their own actions alone. Particularly for each agent, the duality gap convergence criterion in eq. (11) can be understood as a convergence criterion of single-agent constrained RL, and there the proof logic partially follows the analysis of single-agent case [19].

As our Algorithm 1 can adopt any existing algorithm for unconstrained Markov games in the primal update, we consider adopting the V-learning algorithm developed in [31]. This is a decentralized algorithm that incrementally updates the state value functions, which are further used to update the agents' policies via adversarial bandit algorithms. In particular, thanks to the use of state value function, it requires much less memory than other algorithms based on state-action value functions [26]. In Theorem 7 of [31], it is proved that the V-learning algorithm achieves an approximate CE with $\epsilon$ duality gap for any unconstrained Markov game with the state-of-the-art sample complexity $\widetilde{\mathcal{O}}(H^5 S A^2 \epsilon^{-2})$, where $\widetilde{\mathcal{O}}$ hides the logarithm factors and $S = |\mathcal{S}|$, $A = \max_m |\mathcal{A}^{(m)}|$ correspond to the cardinality of the state space and action space, respectively. We then obtain the following sample complexity result.

**Corollary 5.1** (Sample complexity). *Let Theorem 3 hold and assume $r_{0,\max}^{(m)} \geq \frac{1}{H}$. Apply the V-learning algorithm [31] to the primal update of Algorithm 1 and choose $T = \max_m 4\epsilon^{-2}(HR_{\max}^{(m)})^2 \left( \sum_{j=1}^{d_m} (\xi_j^{(m)})^{-2} + HR_{\max}^{(m)} \right)^2$. Then, the output policy $\pi_{\tilde{t}}$ achieves $\max\left( \mathbb{E}[D^{(m)}(\pi_{\tilde{t}})], \mathbb{E}[W^{(m)}(\pi_{\tilde{t}})] \right) \leq 2\epsilon$ with the sample complexity $\widetilde{\mathcal{O}}(H^9 S A^2 \epsilon^{-4})$.*

To the best of our knowledge, this is the first finite-time sample complexity result establsihed for constrained Markov games. Next, we compare our sample complexity with that of two popular RL algorithms. The first one is the V-learning algorithm [31] developed for finding CE of unconstrained Markov games, and it achieves the state-of-the-art sample complexity $\widetilde{\mathcal{O}}(H^5 S A^2 \epsilon^{-2})$. The information theoretical lower bound is $\Omega(H^3 S A \epsilon^{-2})$ [31]. As a comparison, the sample complexity of our primal-dual V-learning algorithm is higher since we require additional projected dual descent steps (12) to account for the constraints. The other one is the basic primal-dual algorithm for solving single-agent constrained RL problems, which uses natural policy gradient in the primal update and achieves the state-of-the-art sample complexity $\mathcal{O}(\epsilon^{-4})$ [7]. Such a complexity has the same dependence on $\epsilon$ as the sample complexity of our algorithm.

## 6 Conclusion

In this work, we propose a surrogate notion of correlated equilibrium (CE) for constrained Markov games and show that it possesses a fundamentally different modification structure from that of CE of Markov game. Moreover, we prove that the corresponding Lagrangian function has zero duality gap. Based on this result, we develop the first primal-dual algorithm that provably converges to CE of constrained Markov games with the convergence rate $\mathcal{O}(\frac{1}{\sqrt{T}})$. Moreover, when adopting the V-learning algorithm as the subroutine in the primal update, our algorithm achieves an approximate CE with $\epsilon$ duality gap with the sample complexity $\mathcal{O}(H^9 S A^2 \epsilon^{-4})$. **Limitations:** In this work, we only develop the basic primal-dual algorithm for constrained Markov games. An interesting future direction is to develop other advanced algorithms such as Nesterov's momentum-accelerated primal-dual algorithms. **Negative social impacts:** This is a fundamental theoretical study and does not have any potential negative social impact.

## Acknowledgment

The work of Z. Chen, S. Ma and Y. Zhou was supported in part by U.S. National Science Foundation under the Grants CCF-2106216 and DMS-2134223.

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
