# Appendix

## Table of Contents

## A   Policy-Induced Joint Measure

We introduce policy-induced joint measure as a useful tool to help prove Theorems 1 and 2. Specifically, for any $h, h'$, we define $p_\pi(s_{1:h}, a_{1:h'})$ as the joint distribution of $(s_{1:h}, a_{1:h'})$ induced by policy $\pi$. Based on the underlying Markov decision process, it can be easily verified that $p_\pi$ takes the following forms.

$$p_\pi(s_1) = \rho(s_1), \tag{16}$$

$$p_\pi(s_{1:h}, a_{1:h-1}) = p_\pi(s_{1:h-1}, a_{1:h-1})\mathcal{P}(s_h|s_{h-1}, a_{h-1}), \tag{17}$$

$$p_\pi(s_{1:h}, a_{1:h}) = \rho(s_1)\pi_1(a_1|s_1) \prod_{h'=2}^{h} \mathcal{P}(s_{h'}|s_{h'-1}, a_{h'-1})\pi_{h'}(a_{h'}|s_{1:h'}, a_{1:h'-1}). \tag{18}$$

Eqs. (17) and (18) further imply that

$$p_\pi(s_{1:h}, a_{1:h}) = p_\pi(s_{1:h}, a_{1:h-1})\pi(a_h|s_{1:h}, a_{1:h-1}). \tag{19}$$

Conversely, given a proper joint probability measure $p$, we can infer its corresponding inducing policy $\pi$ as follows.

**Lemma A.1.** *Consider any joint probability measure $p$ that satisfies eqs.* (16) *and* (17) *(replace all $p_\pi$ with $p$). Then, the following policy $\pi$ induces $p$.*

$$\pi_h(a_h|s_{1:h}, a_{1:h-1}) = \begin{cases} \dfrac{p(s_{1:h}, a_{1:h})}{p(s_{1:h}, a_{1:h-1})}, & \textit{if } p(s_{1:h}, a_{1:h-1}) > 0 \\ \textit{arbitrary distribution}, & \textit{if } p(s_{1:h}, a_{1:h-1}) = 0 \end{cases}. \tag{20}$$

*Proof.* It suffices to prove that $p_\pi(s_{1:h}, a_{1:h}) = p(s_{1:h}, a_{1:h})$ for any $s_{1:h}, a_{1:h}$, i.e., $p$ is exactly the joint measure $p_\pi$ induced by $\pi$. We consider the following two cases.

**Case 1:** $p(s_{1:h}, a_{1:h-1}) > 0$. In this case, we must have that $p(s_{1:h'}, a_{1:h'-1}) > 0$ for any $1 \leq h' \leq h$. Therefore, by eq. (20) we have $\pi_{h'}(a_{h'}|s_{1:h'}, a_{1:h'-1}) = \frac{p(s_{1:h'}, a_{1:h'})}{p(s_{1:h'}, a_{1:h'-1})}$ for any $1 \leq h' \leq h$. Substitute this policy $\pi$ into eq. (18) and note that $\rho(s_1) = p(s_1)$, we obtain that

$$p_\pi(s_{1:h}, a_{1:h}) = p(s_1)\frac{p(s_1, a_1)}{p(s_1)} \prod_{h'=2}^{h} \mathcal{P}(s_{h'}|s_{h'-1}, a_{h'-1})\frac{p(s_{1:h'}, a_{1:h'})}{p(s_{1:h'}, a_{1:h'-1})}$$

$$\overset{(i)}{=} p(s_1, a_1) \prod_{h'=2}^{h} \frac{p(s_{1:h'}, a_{1:h'})}{p(s_{1:h'-1}, a_{1:h'-1})} = p(s_{1:h}, a_{1:h}),$$

where (i) follows from eq. (17) (replace all $p_\pi$ with $p$).

**Case 2:** $p(s_{1:h}, a_{1:h-1}) = 0$. In this case, we have that $p(s_{1:h}, a_{1:h}) = 0$. Hence, it suffices to prove that $p_\pi(s_{1:h}, a_{1:h}) = 0$ as well. We further consider the following two subcases.

(Case 2.1) If $p(s_1) = \rho(s_1) = 0$, then $p_\pi(s_{1:h}, a_{1:h}) = 0$ by substituting $\rho(s_1) = 0$ into eq. (18).

(Case 2.2) If $p(s_1) = \rho(s_1) > 0$ and because Case 2 assumes that $p(s_{1:h}, a_{1:h-1}) = 0$, then there must exist $1 \le h' \le h - 1$ such that $p(s_{1:h'}, a_{1:h'-1}) > 0$ and $p(s_{1:h'+1}, a_{1:h'}) = 0$. On the other hand, note that eq. (18) implies that $p_\pi(s_{1:h}, a_{1:h})$ contains the following multiplicative factor

$$\pi_{h'}(a_{h'} | s_{1:h'}, a_{1:h'-1}) \mathcal{P}(s_{h'+1} | s_{h'}, a_{h'})$$

$$= \mathcal{P}(s_{h'+1} | s_{h'}, a_{h'}) \frac{p(s_{1:h'}, a_{1:h'})}{p(s_{1:h'}, a_{1:h'-1})} \overset{(i)}{=} \frac{p(s_{1:h'+1}, a_{1:h'})}{p(s_{1:h'}, a_{1:h'-1})} = 0 \tag{21}$$

where (i) uses eq. (17). Thus we conclude that $p_\pi(s_{1:h}, a_{1:h}) = 0$. $\qquad\square$

With the policy-induced joint measure $p_\pi$, we can rewrite the value function $V_j^{(m)}(\pi) := \mathbb{E}_\pi \left[ \sum_{h=1}^{H} r_{j,h}^{(m)} \big| s_1 \sim \rho \right]$ and the Lagrangian function eq. (6) as follows.

$$V_j^{(m)}(\pi) = \widetilde{V}_j^{(m)}(p_\pi) := \sum_{s_{1:H}, a_{1:H}} p_\pi(s_{1:H}, a_{1:H}) \sum_{h=1}^{H} r_{j,h}^{(m)}(s_h, a_h), \tag{22}$$

$$L^{(m)}(\pi, \lambda^{(m)}) = \widetilde{L}^{(m)}(p_\pi, \lambda^{(m)}) := \widetilde{V}_0^{(m)}(p_\pi) + \sum_{j=1}^{d_m} \lambda_j \big( \widetilde{V}_j^{(m)}(p_\pi) - c_j^{(m)} \big). \tag{23}$$

Thus, both the value function $V_j^{(m)}(\pi)$ and the Lagrangian function $L^{(m)}(\pi, \lambda^{(m)})$ can be rewritten as linear functions of $p_\pi$. Such a linear form helps simplify the problem and prove the key Theorems 1 and 2.

## B Properties of Modification

In this section, we present some useful properties of the modification operator. Recall that for any policy $\pi$ and any modification operator $\phi^{(m)}$, the modified policy $\phi_h^{(m)} \circ \pi_h$ at time step $h$ is defined as follows: we first generate joint action $a_h = [a_h^{(m)}, a_h^{(\backslash m)}] \sim \pi_h(\cdot | s_{1:h}, a_{1:h-1})$. Then, $\phi_h^{(m)}$ randomly modifies $a_h^{(m)}$ to $\widetilde{a}_h^{(m)} \sim \phi_h^{(m)}(\cdot | s_{1:h}, a_{1:h-1}, a_h^{(m)})$. To summarize, the modified policy $\phi_h^{(m)} \circ \pi_h$ takes the following form.

$$(\phi_h^{(m)} \circ \pi_h)\big( [\widetilde{a}_h^{(m)}, a_h^{(\backslash m)}] | s_{1:h}, a_{1:h-1} \big)$$
$$= \sum_{a_h^{(m)}} \phi_h^{(m)}(\widetilde{a}_h^{(m)} | s_{1:h}, a_{1:h-1}, a_h^{(m)}) \pi_h(a_h | s_{1:h}, a_{1:h-1}). \tag{24}$$

Next, let $p_{\phi^{(m)} \circ \pi}$ and $p_{\widetilde{\phi}^{(m)} \circ \pi}$ be the joint measures induced by the modified policies $\phi^{(m)} \circ \pi$ and $\widetilde{\phi}^{(m)} \circ \pi$, respectively, for any $\phi^{(m)}, \widetilde{\phi}^{(m)}$. In the proof of Theorems 1 and 2, we introduce the following linear combination of these two joint measures.

$$p_\lambda := \lambda p_{\phi^{(m)} \circ \pi} + (1 - \lambda) p_{\widetilde{\phi}^{(m)} \circ \pi}, \lambda \in \mathbb{R}. \tag{25}$$

We note that $\sum_{s_{1:h}, a_{1:h}} p_\lambda(s_{1:h}, a_{1:h}) = 1$ for any $\lambda \in \mathbb{R}$, so $p_\lambda$ is also a proper probability measure if $\lambda$ is selected such that $p_\lambda(s_{1:h}, a_{1:h}) \ge 0$ for any $s_{1:h}, a_{1:h}$. In this case, since the joint measures $p_{\phi^{(m)} \circ \pi}$ and $p_{\widetilde{\phi}^{(m)} \circ \pi}$ satisfy eqs. (16) and (17) by definition, it is easy to verify that $p_\lambda$ also satisfies eqs. (16) and (17) and hence is a proper joint measure. Therefore, by Lemma A.1 we can find its inducing policy $\pi_\lambda$ using eq. (20). Next, we show that such an inducing policy $\pi_\lambda$ can actually be viewed as the policy $\pi$ modified by a certain stochastic modification.

**Lemma B.1.** *Regarding the $p_\lambda$ defined in eq. (25), if $\lambda \in \mathbb{R}$ is selected such that the following two conditions hold for any $s_{1:h}, a_{1:h-1}, a_h^{(m)}, \widetilde{a}_h^{(m)}$:*

$$p_\lambda(s_{1:h}, a_{1:h-1}) \geq 0 \tag{26}$$

$$\lambda p_{\phi^{(m)} \circ \pi}(s_{1:h}, a_{1:h-1}) \phi_h^{(m)}(\widetilde{a}_h^{(m)} | s_{1:h}, a_{1:h-1}, a_h^{(m)})$$
$$+ (1-\lambda) p_{\widetilde{\phi}^{(m)} \circ \pi}(s_{1:h}, a_{1:h-1}) \widetilde{\phi}_h^{(m)}(\widetilde{a}_h^{(m)} | s_{1:h}, a_{1:h-1}, a_h^{(m)}) \geq 0, \tag{27}$$

*then its inducing policy (defined by eq. (20)) can be written as $\pi_\lambda = \phi_\lambda^{(m)} \circ \pi$, where the stochastic modification $\phi_\lambda^{(m)}$ takes the following form.*

$$\phi_{\lambda,h}^{(m)}(\widetilde{a}_h^{(m)} | s_{1:h}, a_{1:h-1}, a_h^{(m)}) =$$
$$\begin{cases} \frac{1}{p_\lambda(s_{1:h}, a_{1:h-1})} \big[ \lambda p_{\phi^{(m)} \circ \pi}(s_{1:h}, a_{1:h-1}) \phi_h^{(m)}(\widetilde{a}_h^{(m)} | s_{1:h}, a_{1:h-1}, a_h^{(m)}) \\ \quad + (1-\lambda) p_{\widetilde{\phi}^{(m)} \circ \pi}(s_{1:h}, a_{1:h-1}) \widetilde{\phi}_h^{(m)}(\widetilde{a}_h^{(m)} | s_{1:h}, a_{1:h-1}, a_h^{(m)}) \big], & \text{if } p_\lambda(s_{1:h}, a_{1:h-1}) > 0 \\ \\ \text{Arbitrary distribution}, & \text{if } p_\lambda(s_{1:h}, a_{1:h-1}) = 0 \end{cases}$$
$$\tag{28}$$

*Proof.* We first show that $\phi_{\lambda,h}^{(m)}(\cdot | s_{1:h}, a_{1:h-1}, a_h^{(m)})$ is a proper stochastic modification. By eq. (28), we only need to consider the case $p_\lambda(s_{1:h}, a_{1:h-1}) > 0$. In this case, based on the condition in eq. (27), we conclude that $\phi_{\lambda,h}^{(m)}(\widetilde{a}_h^{(m)} | s_{1:h}, a_{1:h-1}, a_h^{(m)}) \geq 0$. In addition,

$$\sum_{\widetilde{a}_h^{(m)} \in \mathcal{A}^{(m)}} \phi_{\lambda,h}^{(m)}(\widetilde{a}_h^{(m)} | s_{1:h}, a_{1:h-1}, a_h^{(m)})$$

$$= \frac{1}{p_\lambda(s_{1:h}, a_{1:h-1})} \Big( \lambda p_{\phi^{(m)} \circ \pi}(s_{1:h}, a_{1:h-1}) \sum_{\widetilde{a}_h^{(m)} \in \mathcal{A}^{(m)}} \phi_h^{(m)}(\widetilde{a}_h^{(m)} | s_{1:h}, a_{1:h-1}, a_h^{(m)})$$

$$+ (1-\lambda) p_{\widetilde{\phi}^{(m)} \circ \pi}(s_{1:h}, a_{1:h-1}) \sum_{\widetilde{a}_h^{(m)} \in \mathcal{A}^{(m)}} \widetilde{\phi}_h^{(m)}(\widetilde{a}_h^{(m)} | s_{1:h}, a_{1:h-1}, a_h^{(m)}) \Big)$$

$$= \frac{1}{p_\lambda(s_{1:h}, a_{1:h-1})} \Big( \lambda p_{\phi^{(m)} \circ \pi}(s_{1:h}, a_{1:h-1}) + (1-\lambda) p_{\widetilde{\phi}^{(m)} \circ \pi}(s_{1:h}, a_{1:h-1}) \Big) = 1.$$

Therefore, $\phi_\lambda$ is a proper stochastic modification.

Next, we prove that the policy that induces $p_\lambda$ takes the form $\pi_\lambda = \phi_\lambda^{(m)} \circ \pi$. We consider two cases.

**Case 1:** $p_\lambda(s_{1:h}, a_{1:h-1}) > 0$. In this case, we obtain that

$$\pi_{\lambda,h}(a_h | s_{1:h}, a_{1:h-1})$$

$$\stackrel{(i)}{=} \frac{1}{p_\lambda(s_{1:h}, a_{1:h-1})} \Big( \lambda p_{\phi^{(m)} \circ \pi}(s_{1:h}, a_{1:h}) + (1-\lambda) p_{\widetilde{\phi}^{(m)} \circ \pi}(s_{1:h}, a_{1:h}) \Big)$$

$$\stackrel{(ii)}{=} \frac{1}{p_\lambda(s_{1:h}, a_{1:h-1})} \Big( \lambda p_{\phi^{(m)} \circ \pi}(s_{1:h}, a_{1:h-1}) (\phi_h^{(m)} \circ \pi_h)(a_h | s_{1:h}, a_{1:h-1})$$

$$+ (1-\lambda) p_{\widetilde{\phi}^{(m)} \circ \pi}(s_{1:h}, a_{1:h-1}) (\widetilde{\phi}_h^{(m)} \circ \pi_h)(a_h | s_{1:h}, a_{1:h-1}) \Big)$$

$$\stackrel{(iii)}{=} \frac{1}{p_\lambda(s_{1:h}, a_{1:h-1})} \sum_{\widetilde{a}_h^{(m)}} \Big( \lambda p_{\phi^{(m)} \circ \pi}(s_{1:h}, a_{1:h-1}) \pi_h^{(m)}([\widetilde{a}_h^{(m)}, a_h^{(\backslash m)}] | s_{1:h}, a_{1:h-1})$$

$$\phi_h^{(m)}(a_h^{(m)} | s_{1:h}, a_{1:h-1}, \widetilde{a}_h^{(m)}) + (1-\lambda) p_{\widetilde{\phi}^{(m)} \circ \pi}(s_{1:h}, a_{1:h-1})$$

$$\pi_h^{(m)}([\widetilde{a}_h^{(m)}, a_h^{(\backslash m)}] | s_{1:h}, a_{1:h-1}) \widetilde{\phi}_h^{(m)}(a_h^{(m)} | s_{1:h}, a_{1:h-1}, \widetilde{a}_h^{(m)}) \Big)$$

$$\stackrel{(iv)}{=} \sum_{\widetilde{a}_h^{(m)}} \pi_h^{(m)}([\widetilde{a}_h^{(m)}, a_h^{(\backslash m)}] | s_{1:h}, a_{1:h-1}) \phi_{\lambda,h}^{(m)}(a_h^{(m)} | s_{1:h}, a_{1:h-1}, \widetilde{a}_h^{(m)})$$

where (i) uses eqs. (20) and (25), (ii) uses eq. (19), (iii) uses eq. (24), and (iv) uses eq. (28). This proves the claim due to eq. (24).

**Case 2:** $p_\lambda(s_{1:h}, a_{1:h-1}) = 0$. In this case, both $\phi_{\lambda,h}^{(m)}$ and $\pi_\lambda$ can be arbitrarily defined. Hence, we can simply define $\pi_\lambda$ by eq. (24). $\qquad\square$

## C Proof of Theorem 1

### C.1 Proof of item 1 for unconstrained Markov game

Throughout the proof, for any policy $\pi$, we denote $\widetilde{\phi}^{(m)}$ as the *optimal* stochastic modification associated with $\pi$, i.e., $V_0^{(m)}(\widetilde{\phi}^{(m)} \circ \pi)$ achieves the maximum value over all stochastic modifications. In order for $\pi$ to be a CE, it must satisfy $V_0^{(m)}(\pi) \geq V_0^{(m)}(\widetilde{\phi}^{(m)} \circ \pi)$.

If for any optimal stochastic modification $\widetilde{\phi}^{(m)}$ associated with $\pi$, we can construct a corresponding deterministic modification $\phi^{(m)}$ such that $V_0^{(m)}(\phi^{(m)} \circ \pi) = V_0^{(m)}(\widetilde{\phi}^{(m)} \circ \pi)$, then the condition of item 1 guarantees that $\pi$ is a CE and then item 1 is proved.

Next, for any policy $\pi$ and any associated optimal stochastic modification $\widetilde{\phi}^{(m)}$, we construct a deterministic modification $\phi^{(m)}$ as follows: for any $s_{1:h}, a_{1:h-1}, a_h^{(m)}$, select an arbitrary $\widetilde{a}_h^{(m)}$ such that $\widetilde{\phi}^{(m)}(\widetilde{a}_h^{(m)}|s_{1:h}, a_{1:h-1}, a_h^{(m)}) > 0$ (this always exists) and then simply define $\phi^{(m)}(\widetilde{a}_h^{(m)}|s_{1:h}, a_{1:h-1}, a_h^{(m)}) = 1$, and 0 otherwise. It suffices to prove that $V_0^{(m)}(\phi^{(m)} \circ \pi) = V_0^{(m)}(\widetilde{\phi}^{(m)} \circ \pi)$ for any $\pi$ satisfying the condition of item 1.

To proceed, we claim that one can find $\lambda < 0$ such that the joint measure $p_\lambda := \lambda p_{\phi^{(m)} \circ \pi} + (1 - \lambda) p_{\widetilde{\phi}^{(m)} \circ \pi}$ satisfies eqs. (26) and (27). We will prove the validity of this claim later. Suppose this claim holds. Then based on Lemma B.1, the inducing policy of $p_\lambda$ takes the form $\pi_\lambda = \phi_\lambda^{(m)} \circ \pi$, where $\phi_\lambda^{(m)}$ is defined by eq. (28). Then, we obtain that

$$
\begin{aligned}
V_0^{(m)}(\widetilde{\phi}^{(m)} \circ \pi) &\overset{(i)}{\geq} V_0^{(m)}(\phi_\lambda^{(m)} \circ \pi) \overset{(ii)}{=} \widetilde{V}_0^{(m)}(p_\lambda) \\
&= \widetilde{V}_0^{(m)}\big(\lambda p_{\phi^{(m)} \circ \pi} + (1 - \lambda) p_{\widetilde{\phi}^{(m)} \circ \pi}\big) \\
&\overset{(iii)}{=} \lambda \widetilde{V}_0^{(m)}(p_{\phi^{(m)} \circ \pi}) + (1 - \lambda) \widetilde{V}_0^{(m)}(p_{\widetilde{\phi}^{(m)} \circ \pi}) \\
&\overset{(iv)}{=} \lambda V_0^{(m)}(\phi^{(m)} \circ \pi) + (1 - \lambda) V_0^{(m)}(\widetilde{\phi}^{(m)} \circ \pi),
\end{aligned}
$$

where (i) uses the optimality of $\widetilde{\phi}^{(m)}$, (ii)-(iv) use the linear form of $\widetilde{V}_0^{(m)}(p_\pi)$ defined in eq. (22). The above inequality along with $\lambda < 0$ implies that $V_0^{(m)}(\phi^{(m)} \circ \pi) \geq V_0^{(m)}(\widetilde{\phi}^{(m)} \circ \pi)$. On the other hand, $V_0^{(m)}(\widetilde{\phi}^{(m)} \circ \pi) \geq V_0^{(m)}(\phi^{(m)} \circ \pi)$ based on the optimality of the stochastic modification $\widetilde{\phi}^{(m)}$. Hence, $V_0^{(m)}(\phi^{(m)} \circ \pi) = V_0^{(m)}(\widetilde{\phi}^{(m)} \circ \pi)$ as desired. All left is to find $\lambda < 0$ such that the joint measure $p_\lambda$ satisfies eqs. (26) and (27). We prove them as follows.

**Proof of eq. (26):** Recall that $p_\lambda = \lambda p_{\phi^{(m)} \circ \pi} + (1 - \lambda) p_{\widetilde{\phi}^{(m)} \circ \pi}$ and $\lambda < 0$. If $p_{\phi^{(m)} \circ \pi}(s_{1:h}, a_{1:h-1}) = 0$, it is clear that eq. (26) holds. So we just need to consider the other case where $p_{\phi^{(m)} \circ \pi}(s_{1:h}, a_{1:h-1}) > 0$. In this case and by eq. (18), we must have that $\rho(s_1), (\phi_1^{(m)} \circ \pi_1)(a_1|s_1), \mathcal{P}(s_h|s_{h-1}, a_{h-1}), (\phi_h^{(m)} \circ \pi_h)(a_h|s_{1:h}, a_{1:h-1}) > 0$ for any $h = 2, \ldots, H$. Then, eq. (24) implies that for any $h = 2, \ldots, H$,

$$
\begin{aligned}
0 &< (\phi_h^{(m)} \circ \pi_h)\big(a_h|s_{1:h}, a_{1:h-1}\big) \\
&= \sum_{\widetilde{a}_h^{(m)}} \phi_h^{(m)}(a_h^{(m)}|s_{1:h}, a_{1:h-1}, \widetilde{a}_h^{(m)}) \pi_h\big([\widetilde{a}_h^{(m)}, a_h^{(\backslash m)}]|s_{1:h}, a_{1:h-1}\big).
\end{aligned}
$$

Hence, there must exist $\widehat{a}_h^{(m)}$ such that $\phi_h^{(m)}(a_h^{(m)}|s_{1:h}, a_{1:h-1}, \widehat{a}_h^{(m)}) \pi_h\big([\widehat{a}_h^{(m)}, a_h^{(\backslash m)}]|s_{1:h}, a_{1:h-1}\big) > 0$. As $\phi_h^{(m)}$ is the deterministic modification constructed at the beginning of this proof, we must have

$\phi_h^{(m)}(a_h^{(m)}|s_{1:h}, a_{1:h-1}, \widehat{a}_h^{(m)}) = 1$ and therefore the corresponding stochastic modification satisfies $\widetilde{\phi}_h^{(m)}(a_h^{(m)}|s_{1:h}, a_{1:h-1}, \widehat{a}_h^{(m)}) > 0$. Then, eq. (24) implies that

$$(\widetilde{\phi}_h^{(m)} \circ \pi_h)(a_h|s_{1:h}, a_{1:h-1})$$
$$= \sum_{\widetilde{a}_h^{(m)}} \widetilde{\phi}_h^{(m)}(a_h^{(m)}|s_{1:h}, a_{1:h-1}, \widetilde{a}_h^{(m)})\pi_h([\widetilde{a}_h^{(m)}, a_h^{(\backslash m)}]|s_{1:h}, a_{1:h-1})$$
$$\geq \widetilde{\phi}_h^{(m)}(a_h^{(m)}|s_{1:h}, a_{1:h-1}, \widehat{a}_h^{(m)})\pi_h([\widehat{a}_h^{(m)}, a_h^{(\backslash m)}]|s_{1:h}, a_{1:h-1}) > 0.$$

Similarly, we can prove that $(\widetilde{\phi}_1^{(m)} \circ \pi_1)(a_1|s_1) > 0$ from $(\phi_1^{(m)} \circ \pi_1)(a_1|s_1) > 0$. Therefore, based on eq. (18), it is proved that whenever $p_{\phi^{(m)} \circ \pi}(s_{1:h}, a_{1:h-1}) > 0$, we have

$$p_{\widetilde{\phi}^{(m)} \circ \pi}(s_{1:h}, a_{1:h})$$
$$= \rho(s_1)(\widetilde{\phi}_1^{(m)} \circ \pi_1)(a_1|s_1) \prod_{h'=2}^{h} \mathcal{P}(s_{h'}|s_{h'-1}, a_{h'-1})(\widetilde{\phi}_{h'}^{(m)} \circ \pi_{h'})(a_{h'}|s_{1:h'}, a_{1:h'-1}) > 0. \quad (29)$$

Therefore, eq. (26) holds for

$$0 > \lambda \geq -\frac{p_{\widetilde{\phi}^{(m)} \circ \pi}(s_{1:h}, a_{1:h})}{p_{\phi^{(m)} \circ \pi}(s_{1:h}, a_{1:h}) - p_{\widetilde{\phi}^{(m)} \circ \pi}(s_{1:h}, a_{1:h})} := w(s_{1:h}, a_{1:h}),$$

for any $s_{1:h}, a_{1:h}$ whenever $p_{\phi^{(m)} \circ \pi}(s_{1:h}, a_{1:h}) > p_{\widetilde{\phi}^{(m)} \circ \pi}(s_{1:h}, a_{1:h})$, which implies that $p_{\phi^{(m)} \circ \pi}(s_{1:h}, a_{1:h}) > 0$ and therefore $p_{\widetilde{\phi}^{(m)} \circ \pi}(s_{1:h}, a_{1:h}) > 0$ based on eq. (29). Thus, we conclude that $w(s_{1:h}, a_{1:h}) < 0$. Consider the finite (and possibly empty) set $A_1 := \{w(s_{1:h}, a_{1:h}) : 1 \leq h \leq H, p_{\phi^{(m)} \circ \pi}(s_{1:h}, a_{1:h}) > p_{\widetilde{\phi}^{(m)} \circ \pi}(s_{1:h}, a_{1:h})\}$. If it is non-empty, eq. (26) holds for all $0 > \lambda \geq \max A_1$ for constant $\max A_1 < 0$; Otherwise, eq. (26) holds for all $\lambda < 0$.

**Proof of eq. (27):** If $p_{\phi^{(m)} \circ \pi}(s_{1:h}, a_{1:h-1})\phi_h^{(m)}(\widetilde{a}_h^{(m)}|s_{1:h}, a_{1:h-1}, a_h^{(m)}) = 0$ and $\lambda < 0$, then eq. (27) holds. Consider the other case $p_{\phi^{(m)} \circ \pi}(s_{1:h}, a_{1:h-1})\phi_h^{(m)}(\widetilde{a}_h^{(m)}|s_{1:h}, a_{1:h-1}, a_h^{(m)}) > 0$ and $\lambda < 0$. In this case, we have $p_{\widetilde{\phi}^{(m)} \circ \pi}(s_{1:h}, a_{1:h-1}) > 0$ as proved in the proof of eq. (26), and we also have $\phi_h^{(m)}(\widetilde{a}_h^{(m)}|s_{1:h}, a_{1:h-1}, a_h^{(m)}) = 1 > 0$ and thus $\widetilde{\phi}_h^{(m)}(\widetilde{a}_h^{(m)}|s_{1:h}, a_{1:h-1}, a_h^{(m)}) > 0$ based on the construction of $\phi_h^{(m)}$. Following the same proof logic as that of eq. (26), we can find a constant $A_2 < 0$ such that eq. (27) holds for all $0 > \lambda \geq A_2$.

In summary, we have proved that there exists $\lambda < 0$ that guarantees both eqs. (26) and (27).

## C.2 Proof of item 2 for constrained Markov game

Here we construct a counter example to prove item 2. Consider a constrained Markov game with only one state $\mathcal{S} = \{s\}$, two agents with action spaces $\mathcal{A}_1 = \mathcal{A}_2 = \{0, 1\}$ and horizon $H = 1$. For simplicity, we drop the time step index $h = 1$ and state $s$ in all notations throughout this example. Specifically, we denote $\pi(a^{(1)}, a^{(2)})$, $\pi^{(m)}(a^{(m)})$, $\phi^{(m)}(\widetilde{a}^{(m)}|a^{(m)})$, $m = 1, 2$, as the joint policy, marginal policy and stochastic modification, respectively.

For both agents $m = 1, 2$, we define rewards $r_0^{(m)} = a^{(m)}$, $r_1^{(m)} = a^{(0)} + a^{(1)}$, $r_2^{(m)} = 2 - a^{(0)} - a^{(1)}$ and constraint thresholds $c_1^{(m)} = c_2^{(m)} = 0.6$. Therefore, $V_0^{(m)}(\pi) = \mathbb{E}_\pi r_0^{(m)} = \pi^{(m)}(1)$, $V_1^{(m)}(\pi) = \mathbb{E}_\pi r_1^{(m)} = \pi^{(1)}(1) + \pi^{(2)}(1)$ and $V_2^{(m)}(\pi) = \mathbb{E}_\pi r_2^{(m)} = 2 - \pi^{(1)}(1) - \pi^{(2)}(1) = \pi^{(1)}(0) + \pi^{(2)}(0)$. Therefore, for both agents $m = 1, 2$, their value function constraints $V_1^{(m)}(\pi) \geq 0.6, V_2^{(m)}(\pi) \geq 0.6$ are equivalent to the following condition

$$0.6 \leq \pi^{(1)}(1) + \pi^{(2)}(1) \leq 1.4. \quad (30)$$

Now consider a uniform policy $\overline{\pi}$ where $\overline{\pi}(a^{(1)}, a^{(2)}) = 0.25$ for all $a^{(1)}, a^{(2)} \in \{0, 1\}$. This is a product policy which generates independent uniformly distributed actions $a^{(1)}, a^{(2)}$ with $\overline{\pi}^{(1)}(1) = \overline{\pi}^{(2)}(1) = 0.5$ that satisfy the constraints in eq. (30). Note that $\mathcal{A}^{(1)}$ only includes two actions. Hence, the set of all possible deterministic modifications $\phi^{(1)}$ includes the following three cases.

(i) $\phi^{(1)} \circ \overline{\pi} = \overline{\pi}$: either $\phi^{(1)}$ modifies any $a^{(1)}$ to $a^{(1)}$ or modifies any $a^{(1)}$ to $1 - a^{(1)}$;

(ii) $\phi^{(1)} \circ \overline{\pi} = \pi'$ that always generates $a^{(1)} = 0$ and generates $a^{(2)}$ uniformly at random: $\phi^{(1)}$ modifies any $a^{(1)}$ to 0;

(iii) $\phi^{(1)} \circ \overline{\pi} = \pi''$ that always generates $a^{(1)} = 1$ and generates $a^{(2)}$ uniformly at random: $\phi^{(1)}$ modifies any $a^{(1)}$ to 1.

However, $\pi'$ and $\pi''$ do not satisfy the constraint (30) since $\pi'^{(1)}(1) + \pi'^{(2)}(1) = 0.5$ and $\pi''^{(1)}(1) + \pi''^{(2)}(1) = 1.5$. Hence, the only feasible deterministic modifications $\phi^{(1)}$ are the two ones in (i) with $\phi^{(1)} \circ \overline{\pi} = \overline{\pi}$, which implies that $V_0^{(1)}(\phi^{(1)} \circ \overline{\pi}) = V_0^{(1)}(\overline{\pi}) = \overline{\pi}^{(1)}(1) = 0.5$. Therefore, such a $\overline{\pi}$ satisfies the assumption of item 2.

Now consider a stochastic modification $\phi^{(1)}$ defined by $\phi^{(1)}(1|a_1) = 0.9$ and $\phi^{(1)}(0|a_1) = 0.1$ for $a_1 \in \{0, 1\}$. Then $\phi^{(1)} \circ \overline{\pi}$ independently generates Bernoulli distributed actions $a^{(1)} \sim \mathrm{Bern}(0.9)$ and $a^{(2)} \sim \mathrm{Bern}(0.5)$. Hence, $(\phi^{(1)} \circ \overline{\pi})^{(1)}(1) + (\phi^{(1)} \circ \overline{\pi})^{(2)}(1) = 1.4$, which means $\phi^{(1)}$ is feasible based on eq. (30). In addition, $V_0^{(1)}(\phi^{(1)} \circ \overline{\pi}) = (\phi^{(1)} \circ \overline{\pi})^{(1)}(1) = 0.9$, which is strictly larger than $V_0^{(1)}(\overline{\pi}) = 0.5$. Therefore, $\overline{\pi}$ is not a CE as defined in Definition 3.2.

## D  Proof of Theorem 2

For any policy $\pi$ and its associated joint measure $p_\pi$, recall the following equivalent Lagrangian functions defined in eq. (23).

$$L^{(m)}(\pi, \lambda^{(m)}) = \widetilde{L}^{(m)}(p_\pi, \lambda^{(m)}).$$

Then, the desired strong duality result shown in eq. (7) is equivalent to the following equation.

$$\max_{p \in \mathcal{X}} \min_{\lambda^{(m)} \in \mathbb{R}_+^{d_m}} \widetilde{L}^{(m)}(p, \lambda^{(m)}) = \min_{\lambda^{(m)} \in \mathbb{R}_+^{d_m}} \max_{p \in \mathcal{X}} \widetilde{L}^{(m)}(p, \lambda^{(m)}),$$

where the set $\mathcal{X} := \{p_{\phi^{(m)} \circ \pi} : \phi^{(m)} \text{ is a stochastic modification}\}$ is defined for the fixed $\pi$. The nice property of the Lagrangian function $\widetilde{L}^{(m)}(p, \lambda^{(m)})$ is that it is a linear function in $p$, which has an advantage toward establishing strong duality.

Based on the minimax theorem (Lemma 9.2 of [2]), it suffices to prove the following properties:

(I).  $\widetilde{L}^{(m)}(p, \cdot)$ is convex and lower semi-continuous, and $\widetilde{L}^{(m)}(\cdot, p)$ is concave. These properties directly follow from the definition of $\widetilde{L}$ in eq. (23).

(II).  $\mathbb{R}_+^{d_m}$ is a convex set, which holds obviously.

(III).  $\mathcal{X}$ is a convex set, which follows from Lemma B.1 since eqs. (26) and (27) always hold for $\lambda \in [0, 1]$.

(IV).  $\mathcal{X}$ is a compact set.

Hence, it remains to prove (IV).

As the state space $\mathcal{S}$, action apace $\mathcal{A}$ and the horizon $H$ are finite, we can represent $p_\pi$ as a vector with entries $p_\pi(s_{1:H}, a_{1:H})$ for every $s_{1:H}, a_{1:H} \in \mathcal{S}^H \times \mathcal{A}^H$. Hence, the set $\mathcal{X} \subset [0, 1]^{(|\mathcal{S}||\mathcal{A}|)^H}$ is bounded. Then, it suffices to prove that $\mathcal{X}$ is a closed set, i.e., $p \in \mathcal{X}$ if $p_{\phi_{[k]}^{(m)} \circ \pi}(s_{1:H}, a_{1:H}) \xrightarrow{k} p(s_{1:H}, a_{1:H}), \forall s_{1:H}, a_{1:H}$ for some $p_{\phi_{[k]}^{(m)} \circ \pi} \in \mathcal{X}$ (Note that the notation $\phi_{[k]}^{(m)}$ indexed by $k$ differs from $\phi_h^{(m)}$ where $h$ denotes time step).

Similar to $\mathcal{X}$, any stochastic modification $\phi^{(m)}$ can also be seen as a bounded finite-dimensional vector with entries $\phi^{(m)}(\widetilde{a}_h^{(m)}|s_{1:h}, a_{1:h-1}, a_h^{(m)}) \in [0, 1]$. Hence, $\{\phi_{[k]}^{(m)} : k \in \mathbb{N}^+\}$ has a convergent sub-sequence $\{\phi_{[k_i]}^{(m)} : i \in \mathbb{N}^+\}$ such that $\phi_{[k_i]}^{(m)}(\widetilde{a}_h^{(m)}|s_{1:h}, a_{1:h-1}, a_h^{(m)}) \xrightarrow{i} \phi^*(\widetilde{a}_h^{(m)}|s_{1:h}, a_{1:h-1}, a_h^{(m)})$ for any $s_{1:h}, a_{1:h-1}, a_h^{(m)}, \widetilde{a}_h^{(m)}$, which implies that $\phi^*(\widetilde{a}_h^{(m)}|s_{1:h}, a_{1:h-1}, a_h^{(m)}) \geq 0$ and $\sum_{\widetilde{a}_h^{(m)}} \phi^*(\widetilde{a}_h^{(m)}|s_{1:h}, a_{1:h-1}, a_h^{(m)}) = 1$. Therefore, $\phi^*$ is a proper stochastic modification.

Then based on eq. (24), it holds for any $s_{1:h}, a_{1:h}$ that

$$
\begin{aligned}
&(\phi_{[k_i],h}^{(m)} \circ \pi_h)\big(a_h|s_{1:h}, a_{1:h-1}\big) \\
&= \sum_{\widetilde{a}_h^{(m)}} \phi_{[k_i],h}^{(m)}(a_h^{(m)}|s_{1:h}, a_{1:h-1}, \widetilde{a}_h^{(m)})\pi_h\big([\widetilde{a}_h^{(m)}, a_h^{(\backslash m)}]|s_{1:h}, a_{1:h-1}\big) \\
&\xrightarrow{i} \sum_{\widetilde{a}_h^{(m)}} \phi_h^*(a_h^{(m)}|s_{1:h}, a_{1:h-1}, \widetilde{a}_h^{(m)})\pi_h\big([\widetilde{a}_h^{(m)}, a_h^{(\backslash m)}]|s_{1:h}, a_{1:h-1}\big) \\
&= (\phi_h^* \circ \pi_h)(a_h|s_{1:h}, a_{1:h-1}). 
\end{aligned} \tag{31}
$$

On one hand, the above inequality and eq. (18) imply that for any $s_{1:h}, a_{1:h}$, $p_{\phi_{[k_i]}^{(m)} \circ \pi}(s_{1:h}, a_{1:h}) \xrightarrow{i}$ $p_{\phi^* \circ \pi}(s_{1:h}, a_{1:h})$. On the other hand, $p_{\phi_{[k_i]}^{(m)} \circ \pi}(s_{1:h}, a_{1:h}) \xrightarrow{i} p(s_{1:h}, a_{1:h})$. Therefore, $p = p_{\phi^* \circ \pi}$ for $\phi^*$ being a stochastic modification, and thus $p \in \mathcal{X}$.

## E  The Range of the Optimal Dual Variable

Before proving Theorem 3 and Corollary 5.1 on the non-asymptotic convergence of Algorithm 1, we first consider the optimal dual variable $\lambda_*^{(m)}$ of the minimax optimization problem in eq. (7) and derive its range below, which is important for the selection of the projection set $\Lambda^{(m)}$ in Algorithm 1.

**Lemma E.1.** *The optimal dual variable $\lambda_*^{(m)}$ satisfies the following range.*

$$
\lambda_{*,j}^{(m)} \leq \frac{Hr_{0,\max}^{(m)}}{\xi_j^{(m)}}, j = 1, \ldots, d_m. \tag{32}
$$

*Proof.* Given $\pi$, denote $\phi_*^{(m)}$ as the optimal solution to the constrained optimization problem in eq. (4) and denote $\widetilde{\phi}^{(m)}$ as the stochastic modification that satisfies Assumption 1, i.e., $V^{(m)}(\widetilde{\phi}^{(m)} \circ \pi) - c^{(m)} \geq \xi^{(m)}$. Then we have

$$
\begin{aligned}
Hr_{0,\max}^{(m)} &\overset{(i)}{\geq} V_0^{(m)}\big(\phi_*^{(m)} \circ \pi\big) \\
&\overset{(ii)}{=} \max_{\phi^{(m)}} L^{(m)}\big(\phi^{(m)} \circ \pi, \lambda_*^{(m)}\big) \\
&\geq L^{(m)}\big(\widetilde{\phi}^{(m)} \circ \pi, \lambda_*^{(m)}\big) \\
&= V_0^{(m)}(\widetilde{\phi}^{(m)} \circ \pi) + \sum_{j=1}^{d_m} \lambda_{*,j}^{(m)}\big(V_j^{(m)}(\widetilde{\phi}^{(m)} \circ \pi) - c_j^{(m)}\big) \\
&\overset{(iii)}{\geq} \sum_{j=1}^{d_m} \lambda_{*,j}^{(m)} \xi_j^{(m)},
\end{aligned}
$$

where (i) and (iii) use $V_0^{(m)}(\pi) \in [0, Hr_{0,\max}^{(m)}], \forall j = 0, 1, \ldots, d_m$ which is directly implied by Assumption 2, (ii) uses Theorem 2 which implies the equivalence between the constrained optimization problem in eq. (4) and the minimax optimization problem in eq. (7), and (iii) also uses $\lambda_{*,j}^{(m)} \geq 0$ and $V_j^{(m)}(\pi) - c_j^{(m)} \geq \xi_j^{(m)}$. Since $\xi_j^{(m)} > 0$, the above inequality implies eq. (32)  $\qquad \square$

## F  Proof of Theorem 3

Assumption 2 and the value functions defined in eq. (4) imply that for any $m = 1, \ldots, M, j = 0, 1, \ldots, d_m$ and joint policy $\pi$, we have

$$
0 \leq V_j^{(m)}(\pi) = \mathbb{E}_\pi\Big[\sum_{h=1}^{H} r_{j,h}^{(m)}(s_h, a_h)\Big|s_1 \sim \rho\Big] \leq Hr_{j,\max}^{(m)}. \tag{33}
$$

Hence, for any $m = 1, \ldots, M$ and joint policy $\pi$

$$\|V^{(m)}(\pi)\| = \sqrt{\sum_{j=1}^{d_m} V_j^{(m)}(\pi)^2} \leq H \sqrt{\sum_{j=1}^{d_m} (r_{j,\max}^{(m)})^2} = H R_{\max}^{(m)} \tag{34}$$

Furthermore, Assumption 1 implies that there is a joint policy $\pi'$ such that $0 \leq c^{(m)} \leq V^{(m)}(\pi')$, so

$$\|c^{(m)}\| \leq \|V^{(m)}(\pi')\| \leq H R_{\max}^{(m)}. \tag{35}$$

Then,

$$
\begin{aligned}
0 &\leq \|\lambda_T^{(m)}\|^2 \\
&\stackrel{(i)}{=} \sum_{t=0}^{T-1} \left( \|\lambda_{t+1}^{(m)}\|^2 - \|\lambda_t^{(m)}\|^2 \right) \\
&\stackrel{(ii)}{\leq} \sum_{t=0}^{T-1} \left( \left\| \lambda_t^{(m)} - \eta \big(V^{(m)}(\pi_t) - c^{(m)}\big) \right\|^2 - \|\lambda_t^{(m)}\|^2 \right) \\
&\stackrel{(iii)}{\leq} 2\eta \sum_{t=0}^{T-1} \lambda_t^{(m)\top} \big(c^{(m)} - V^{(m)}(\pi_t)\big) + \eta^2 \sum_{t=0}^{T-1} \big( \|V^{(m)}(\pi_t)\| + \|c^{(m)}\| \big)^2 \\
&\stackrel{(iv)}{\leq} 2\eta \sum_{t=0}^{T-1} \lambda_t^{(m)\top} \big( V^{(m)}(\phi_{t*}^{(m)} \circ \pi_t) - V^{(m)}(\pi_t) \big) + 4T(\eta H R_{\max}^{(m)})^2,
\end{aligned}
$$

where (i) uses the initialization $\lambda_0^{(m)} = 0$, (ii) uses eq. (12) and $0 \in \Lambda^{(m)}$, (iii) uses triangular inequality, and (iv) uses eqs. (34) and (35) and the constraint that $V^{(m)}(\phi_{t*}^{(m)} \circ \pi) \geq c^{(m)}$ satisfied by the optimal modification $\phi_{t*}^{(m)}$ of the constrained optimization problem in eq. (4) for $\pi = \pi_t$. Rearranging the above inequality yields that

$$\sum_{t=0}^{T-1} \lambda_t^{(m)\top} \big( V^{(m)}(\pi_t) - V^{(m)}(\phi_t^{(m)} \circ \pi_t) \big) \leq 2\eta T (H R_{\max}^{(m)})^2. \tag{36}$$

Note that

$$
\begin{aligned}
0 &\leq \sum_{t=0}^{T-1} \left( \max_{\phi^{(m)}} L^{(m)}\big(\phi^{(m)} \circ \pi_t, \lambda_t^{(m)}\big) - L^{(m)}\big(\phi_{t*}^{(m)} \circ \pi_t, \lambda_t^{(m)}\big) \right) \\
&\stackrel{(i)}{=} \sum_{t=0}^{T-1} \left( \max_{\phi^{(m)}} V_{\lambda_t}^{(m)}\big(\phi^{(m)} \circ \pi_t\big) - V_{\lambda_t}^{(m)}\big(\phi_{t*}^{(m)} \circ \pi_t\big) \right) \\
&\stackrel{(ii)}{\leq} \sum_{t=0}^{T-1} \left( \epsilon + V_{\lambda_t}^{(m)}(\pi_t) - V_{\lambda_t}^{(m)}\big(\phi_{t*}^{(m)} \circ \pi_t\big) \right) \\
&\stackrel{(iii)}{=} \sum_{t=0}^{T-1} \left( \epsilon + V_0^{(m)}(\pi_t) - V_0^{(m)}\big(\phi_{t*}^{(m)} \circ \pi_t\big) + \lambda_t^{(m)\top} \big( V^{(m)}(\pi_t) - V^{(m)}(\phi_{t*}^{(m)} \circ \pi_t) \big) \right) \quad (37) \\
&\stackrel{(iv)}{\leq} \sum_{t=0}^{T-1} \left( \epsilon - D^{(m)}(\pi_t) \right) + 2\eta T (H R_{\max}^{(m)})^2,
\end{aligned}
$$

where (i) uses the rewritten Lagrangian function $L^{(m)}(\phi^{(m)} \circ \pi, \lambda^{(m)}) = V_\lambda^{(m)}(\phi^{(m)} \circ \pi) - \lambda^{(m)\top} c^{(m)}$, (ii) uses eq. (11), (iii) uses $V_\lambda^{(m)}(\pi) = V_0^{(m)}(\pi) + \lambda^{(m)\top} V^{(m)}(\pi), \forall \pi$ implies by eqs. (2) and (9), and (iv) uses eqs. (5) and (36). Rearranging the above inequality yields that

$$\mathbb{E}_{\widehat{t}}\big[D^{(m)}(\pi_t)\big] = \frac{1}{T} \sum_{t=0}^{T-1} D^{(m)}(\pi_t)) \leq 2\eta (H R_{\max}^{(m)})^2 + \epsilon,$$

which proves the duality gap in eq. (14) by substituting $\eta = \frac{1}{\sqrt{T}}$.

Next, we prove the constraint violation in eq. (15).

For any $\lambda^{(m)} \in \Lambda^{(m)}$, it holds that

$$
\begin{aligned}
&\|\lambda_{t+1}^{(m)} - \lambda^{(m)}\|^2 \\
&\overset{(i)}{\leq} \|\lambda_t^{(m)} - \eta\big(V^{(m)}(\pi_t) - c^{(m)}\big) - \lambda^{(m)}\|^2 \\
&\overset{(ii)}{\leq} \|\lambda_t^{(m)} - \lambda^{(m)}\|^2 - 2\eta(\lambda_t^{(m)} - \lambda^{(m)})^\top \big(V^{(m)}(\pi_t) - c^{(m)}\big) + \eta^2 \big(\|V^{(m)}(\pi_t)\| + \|c^{(m)}\|\big)^2 \\
&\overset{(iii)}{\leq} \|\lambda_t^{(m)} - \lambda^{(m)}\|^2 - 2\eta(\lambda_t^{(m)} - \lambda^{(m)})^\top \big(V^{(m)}(\pi_t) - c^{(m)}\big) + 4(\eta H R_{\max}^{(m)})^2
\end{aligned}
$$

where (i) uses eq. (12) and $\lambda^{(m)} \in \Lambda^{(m)}$, (ii) uses triangular inequality, (iii) uses eqs. (34) and (35). Telescoping the above inequality over $t = 0, 1, \ldots, T-1$ and using $\lambda_0^{(m)} = 0$ yields that

$$
\eta \sum_{t=0}^{T-1} (\lambda_t^{(m)} - \lambda^{(m)})^\top \big(V^{(m)}(\pi_t) - c^{(m)}\big) \leq \frac{1}{2}\|\lambda^{(m)}\|^2 + 2T(\eta H R_{\max}^{(m)})^2. \tag{38}
$$

Since $V^{(m)}(\phi_{t*}^{(m)} \circ \pi_t) \geq c^{(m)}$ and $\lambda_t^{(m)} \in \mathbb{R}_+^{d_m}$, eq. (37) implies that

$$
\eta \sum_{t=0}^{T-1} \lambda_t^{(m)\top} \big(c^{(m)} - V^{(m)}(\pi_t)\big) \leq \eta \sum_{t=0}^{T-1} \big(\epsilon + V_0^{(m)}(\pi_t) - V_0^{(m)}(\phi_{t*}^{(m)} \circ \pi)\big) \tag{39}
$$

Summing up eqs. (38) and (39) yields that

$$
\begin{aligned}
\eta \sum_{t=0}^{T-1} \lambda^{(m)\top} \big(c^{(m)} - V^{(m)}(\pi_t)\big) \leq &\eta \sum_{t=0}^{T-1} \big(\epsilon + V_0^{(m)}(\pi_t) - V_0^{(m)}(\phi_{t*}^{(m)} \circ \pi)\big) \\
&+ \frac{1}{2}\|\lambda^{(m)}\|^2 + 2T(\eta H R_{\max}^{(m)})^2.
\end{aligned} \tag{40}
$$

Denote $\Phi_t^{(m)} := \big\{\phi^{(m)} : V^{(m)}(\phi^{(m)} \circ \pi_t) \geq \min\big(c^{(m)}, V^{(m)}(\pi_t)\big)\big\}$, which is a non-empty set that includes identity modification $\phi^{(m)}$ such that $I^{(m)} \circ \pi_t = \pi_t$. Hence,

$$
\begin{aligned}
V_0^{(m)}(\phi_{t*}^{(m)} \circ \pi_t) &= \max_{\phi^{(m)}} \min_{\lambda^{(m)} \in \mathbb{R}_+^{d_m}} L^{(m)}(\phi^{(m)} \circ \pi_t, \lambda^{(m)}) \\
&\overset{(i)}{=} \min_{\lambda^{(m)} \in \mathbb{R}_+^{d_m}} \max_{\phi^{(m)}} L^{(m)}(\phi^{(m)} \circ \pi_t, \lambda^{(m)}) \\
&\overset{(ii)}{\geq} \max_{\phi^{(m)} \in \Phi_t^{(m)}} L^{(m)}\big(\phi^{(m)} \circ \pi_t, \lambda_{t*}^{(m)}\big) \\
&= \max_{\phi^{(m)} \in \Phi_t^{(m)}} \big(V_0^{(m)}\big(\phi^{(m)} \circ \pi_t\big) + (\lambda_{t*}^{(m)})^\top \big[V^{(m)}\big(\phi^{(m)} \circ \pi_t\big) - c^{(m)}\big]\big) \\
&\overset{(iii)}{\geq} \max_{\phi^{(m)} \in \Phi_t^{(m)}} V_0^{(m)}\big(\phi^{(m)} \circ \pi_t\big) + (\lambda_{t*}^{(m)})^\top \min\big(0, V^{(m)}(\pi_t) - c^{(m)}\big) \\
&\overset{(iv)}{\geq} V_0^{(m)}(\pi_t) - (\lambda_{t*}^{(m)})^\top \big(c^{(m)} - V^{(m)}(\pi_t)\big)_+
\end{aligned}
$$

where (i) uses Theorem 2, (ii) uses the fact that $\Phi_t^{(m)}$ is only a subset of stochastic modifications and denotes that $\lambda_{t*}^{(m)} = \arg\min_{\lambda^{(m)} \in \mathbb{R}_+^{d_m}} \max_{\phi^{(m)}} L^{(m)}(\phi^{(m)} \circ \pi_t, \lambda^{(m)})$, (iii) uses $\lambda_{t*}^{(m)} \in \mathbb{R}_+^{d_m}$ and the definition of $\Phi_t^{(m)}$, and (iv) uses the fact that the identity modification $\phi^{(m)} \in \Phi_t^{(m)}$. Substituting the above inequality into eq. (40) and rearranging it, we obtain that

$$
\eta \sum_{t=0}^{T-1} \Big(\lambda^{(m)\top}\big(c^{(m)} - V^{(m)}(\pi_t)\big) - (\lambda_{t*}^{(m)})^\top \big(c^{(m)} - V^{(m)}(\pi_t)\big)_+\Big)
$$

$$\leq \frac{1}{2}\|\lambda^{(m)}\|^2 + 2T(\eta HR^{(m)}_{\max})^2 + \eta T\epsilon. \tag{41}$$

Using eq. (32) and selecting $\lambda^{(m)}_j = \frac{2Hr^{(m)}_{0,\max}}{\xi^{(m)}_j}\mathbb{1}\{V^{(m)}_j(\pi_t) \leq c^{(m)}_j\}$ (this satisfies $\lambda^{(m)} \in \Lambda^{(m)}$), we obtain that

$$\lambda^{(m)\top}\big(c^{(m)} - V^{(m)}(\pi_t)\big) - (\lambda^{(m)}_{t*})^\top\big(c^{(m)} - V^{(m)}(\pi_t)\big)_+$$
$$\geq \sum_{j=1}^{d_m} \frac{Hr^{(m)}_{0,\max}}{\xi^{(m)}_j}\big(c^{(m)}_j - V^{(m)}_j(\pi_t)\big)_+,$$

where the last inequality uses eq. (32). Substituting the above inequality into eq. (41) yields that

$$\eta Hr^{(m)}_{0,\max}\sum_{t=0}^{T-1}\sum_{j=1}^{d_m}(\xi^{(m)}_j)^{-1}\big(c^{(m)}_j - V^{(m)}_j(\pi_t)\big)_+$$
$$\leq \frac{1}{2}\|\lambda^{(m)}\|^2 + 2T(\eta HR^{(m)}_{\max})^2 + \eta T\epsilon$$
$$\overset{(i)}{\leq} 2(Hr^{(m)}_{0,\max})^2\sum_{j=1}^{d_m}(\xi^{(m)}_j)^{-2} + 2T(\eta HR^{(m)}_{\max})^2 + \eta T\epsilon,$$

where (i) uses $\|\lambda^{(m)}\| \leq 2Hr^{(m)}_{0,\max}\sqrt{\sum_{j=1}^{d_m}(\xi^{(m)}_j)^{-2}}$ for our choice $\lambda^{(m)}_j = \frac{2Hr^{(m)}_{0,\max}}{\xi^{(m)}_j}$
$\mathbb{1}\{V^{(m)}_j(\pi_t) \leq c^{(m)}_j\}$. Dividing both sides of the above inequality by $\eta THr^{(m)}_{0,\max}$ and substituting $\eta = \frac{1}{\sqrt{T}}$, we prove the constraint violation in eq. (15).

# G   Proof of Corollary 5.1

The surrogate rewards defined in eq. (8) has the following bound

$$0 \leq R^{(m)}_{\lambda_t,h}(s_h, a_h) = r^{(m)}_{0,h}(s_h, a_h) + \lambda^{(m)\top}_t r^{(m)}_h(s_h, a_h)$$
$$\leq r^{(m)}_{0,h}(s_h, a_h) + \|\lambda^{(m)}_t\|\|r^{(m)}_h(s_h, a_h)\|$$
$$\overset{(i)}{\leq} r^{(m)}_{0,\max} + 2Hr^{(m)}_{0,\max}R^{(m)}_{\max}\sqrt{\sum_{j=1}^{d_m}(\xi^{(m)}_j)^{-2}} := \widetilde{R}^{(m)}_{\max} \tag{42}$$

where (i) uses Assumption 2 and $\lambda^{(m)}_{t,j} \in \big[0, \frac{2Hr^{(m)}_{0,\max}}{\xi^{(m)}_j}\big]$ (since $\lambda^{(m)}_t \in \Lambda^{(m)}$ based on eq. (12)). Note that the V-learning in [31] assumes the rewards to range in $[0, 1]$. To adjust to this assumption, we apply V-learning to the scaled rewards $\frac{1}{\widetilde{R}^{(m)}_{\max}}R^{(m)}_{\lambda_t,h}(s_h, a_h) \in [0, 1]$ with corresponding value function $\frac{1}{\widetilde{R}^{(m)}_{\max}}V^{(m)}_{\lambda_t}$. Then based on Theorem 7 of [31], it takes $\widetilde{\mathcal{O}}(H^5SA^2(\epsilon/\widetilde{R}^{(m)}_{\max})^{-2}) = \widetilde{\mathcal{O}}(H^5SA^2\epsilon^{-2})$ samples to reach the $\epsilon/\widetilde{R}^{(m)}_{\max}$-CE of this scaled Markov game with probability at least $1 - \delta/T$ for any $\delta \in (0, 1)$ (we replace $\delta$ with $\delta/T$ which only changes the hidden logarithm factor in $\widetilde{O}$), that is,

$$\max_{\phi^{(m)}}\frac{1}{\widetilde{R}^{(m)}_{\max}}V^{(m)}_\lambda(\phi^{(m)} \circ \pi_t) - \frac{1}{\widetilde{R}^{(m)}_{\max}}V^{(m)}_\lambda(\pi_t) \leq \frac{\epsilon}{\widetilde{R}^{(m)}_{\max}},$$

which is equivalent to eq. (11). Applying union bound over the $T$ iterations yields that eq. (11) holds for all iterations $t = 0, 1, \ldots, T - 1$ with probability at least $1 - \delta$. In that case, the convergence rates in eqs. (14) and (15) hold. Substituting $T = \max_m 4\epsilon^{-2}(HR^{(m)}_{\max})^2\big(\sum_{j=1}^{d_m}(\xi^{(m)}_j)^{-2} + HR^{(m)}_{\max}\big)^2$ and $r^{(m)}_{0,\max} \geq \frac{1}{H}$ into these convergence rates yields that

$$\mathbb{E}_{\widetilde{t}}\big(D^{(m)}(\pi_{\widetilde{t}})\big) \leq \frac{2(HR^{(m)}_{\max})^2}{\sqrt{T}} + \epsilon \leq 2\epsilon,$$

$$\mathbb{E}_{\widehat{t}}\big(W^{(m)}(\pi_{\widehat{t}})\big) \le \frac{2HR_{\max}^{(m)}}{\sqrt{T}} \sum_{j=1}^{d_m}(\xi_j^{(m)})^{-2} + \frac{2H(R_{\max}^{(m)})^2}{r_{0,\max}^{(m)}\sqrt{T}} + \frac{\epsilon}{Hr_{0,\max}^{(m)}}$$

$$\le \frac{1}{\sqrt{T}}\Big(2HR_{\max}^{(m)}\sum_{j=1}^{d_m}(\xi_j^{(m)})^{-2} + 2(HR_{\max}^{(m)})^2\Big) + \epsilon \le 2\epsilon.$$

The above two inequalities prove that $\max\big(\mathbb{E}_{\widehat{t}}D^{(m)}(\pi_{\widehat{t}}), \mathbb{E}_{\widehat{t}}W^{(m)}(\pi_{\widehat{t}})\big) \le 2\epsilon$.

Since each of the $T = \mathcal{O}(H^4\epsilon^{-2})$ iterations takes $\widetilde{\mathcal{O}}(H^5SA^2\epsilon^{-2})$ samples, the required sample complexity is $T\widetilde{\mathcal{O}}(H^5SA^2\epsilon^{-2}) = \widetilde{\mathcal{O}}(H^9SA^2\epsilon^{-4})$.