# OpenReview forum: "Finding Correlated Equilibrium of Constrained Markov Game: A Primal-Dual Approach"
_NeurIPS.cc/2022/Conference — NeurIPS 2022 Accept_

### Official Review · Reviewer_ddwR · 2022-07-10

**Rating:** 7
**Confidence:** 4
**Soundness:** 4 excellent
**Presentation:** 4 excellent
**Contribution:** 3 good

**Summary:**

This paper examines Constrained Markov games.
It proposes:
(1) a notion of correlated equilibrium for Markov games
demonstrating the shortcoming of a deterministic modification function

(2) under the assumption of the satisfaction of Slater condition, a zero-duality gap is proven for CE in Markov games in finite-horizon with a non-deterministic modification function

(3) the problem of finding a CE policy for each agent is written as a mathematical program with appropriate inequality constraints and the modification function as variables. After the Lagrangian function has been formed, the Lagrangian is written as a function of probability measures of the episodes induced by applying the modification function on the policy. It then follows that the Lagrangian is convex w.r.t to these variables.

(4) from this point forward, a a primal-dual optimization algorithm can be applied. a part of the "primal" loop can be interpreted as the problem of finding a CE in an unconstrained Markov game which allows



**Questions:**

Can the authors argue in what settings the authors results carry over to Nash equilibria? I know this is  hard question, I guess it is interesting for future consideration.

**Limitations:**

This is theoretical work

**Strengths And Weaknesses:**

Overall the paper is well-written and I think it is above the acceptance bar.

The main strength are:
- The result about non-asymptotic convergence (they provide rates which is really interesting).
- The above allows the authors to talk about sample complexity.

---

> ### Author Response · Authors · 2022-07-31
> **Authors' reply to ddwR**
>
> Thank you very much for reviewing our manuscript and providing valuable feedback. Below is a response to the review comments. We have submitted a revised version with all revisions marked in **"red"**. Please let us know if further clarifications are needed.
>
> Q: In what settings can we carry over the results to Nash equilibrium (NE)?
>
> A: Great question. Our results can be carried over to NE by replacing the subroutine algorithm for finding CE of the unconstrained Markov game in eq. (11) with an algorithm that finds NE of the same unconstrained Markov game, and most of the proof logic for convergence analysis remains the same. However, the resulting sample complexity will be very high since finding NE for general unconstrained Markov game is a PPAD-complete problem.
>
> To make the sample complexity tractable for finding NE, the surrogate reward $R_{\lambda}^{(m)}(s_h,a_h):=r_0^{(m)}(s_h,a_h)+\lambda^{(m)\top} r^{(m)}(s_h,a_h)$ of the unconstrained Markov game should have certain structures, e.g., the zero-sum structure [1] or the potential structure [2]. The zero-sum structure cannot be satisfied in general as there can be more than two players and the surrogate reward $R_{\lambda}^{(m)}$ involves the dual variable, whereas the potential structure is satisfied if and only if the following two conditions hold.
>
> (1) The constraints-related value function  $V_{j}^{(m)}(\pi^{(m)}\times\pi^{(\backslash m)})$ ($j=1,\ldots,d_m$) for any product policy $\pi^{(m)}\times\pi^{(\backslash m)}$ does not rely on $\pi^{(m)}$.
>
> (2) There exists a potential function $\Phi(\pi)\in\mathbb{R}$ such that the objective-related value functions satisfy the following potential function assumption for all $m$, $\pi^{(m)}\times\pi^{(\backslash m)}$ and $\widetilde{\pi}^{(m)}\times\pi^{(\backslash m)}$.
>
> $$V_0^{(m)}(\widetilde{\pi}^{(m)}\times\pi^{(\backslash m)})-V_0^{(m)}(\pi^{(m)}\times\pi^{(\backslash m)})=\Phi(\widetilde{\pi}^{(m)}\times\pi^{(\backslash m)})-\Phi(\pi^{(m)}\times\pi^{(\backslash m)}).$$
>
> [1] Y. Zhao, Y. Tian, J. D. Lee, and S. S. Du. Provably efficient policy gradient methods for two-player zero-sum markov games. ArXiv:2102.08903, 2021.
>
> [2] S. Leonardos, W. Overman, I. Panageas, and G. Piliouras. Global convergence of multi-agent policy gradient in markov potential games. ArXiv:2106.01969, 2021.

---

> > ### Comment · Reviewer_ddwR · 2022-08-07
> > **Thank you for the response**
> >
> > I would like to thank the authors for their response. I feel the paper is above the acceptance bar,  I cannot increase my score further though.
> > I will continue the discussion with fellow reviewers and AC.

---

> > > ### Author Response · Authors · 2022-08-08
> > > **Follow-up response to reviewer ddwR**
> > >
> > > Thank you very much for the follow-up response and appreciating our work.

---

### Official Review · Reviewer_jkm3 · 2022-07-11

**Rating:** 6
**Confidence:** 2
**Soundness:** 3 good
**Presentation:** 3 good
**Contribution:** 2 fair

**Summary:**

This paper proposes a primal-dual algorithm for solving for correlated equilibria (CE) in Constrained markov games. This can be seen either as (a) an extension of (nash) constrained markov games (or MDPs) to CE, or (b) adding constraints to an unconstrained CE for Markov games. The primary technical contribution is in Theorem 2, which shows strong duality and points towards a primal dual algorithm.The authors also study convergence rates and sample complexity.


**Questions:**

1. My understanding is that the existence of a constrained CE may not exist. Is it true that a constrained Nash exists, a constrained CE would as well. Furthermore, are there cases where a constrained CE exists, but not for constrained Nash?

2. The authors introduce CE as a surrogate for Nash. However, amongst many game theorists, CE is in and of itself an equilibrium concept that is of interest, and involves a trusted mediator giving recommendations to players (or cryptographic protocols). Most frequently, one can maximize social welfare while maintaining incentive compatibility. I would think that one can easily force this objective in, by adding lower bounds on social welfare as a constraint for each player while performing binary search on social welfare. Is there a cleaner way of selecting between various constrained CE?

3. Again, related to the classical definition of a CE (i.e., with a mediator). The setting here is one with finite horizon. Given that the authors allow for nonstationary strategies, the game can technically be unrolled (albeit incurring exponential blowup) into a tree. From this perspective, the concept of a CE is actually very similar to that of an extensive-form correlated equilibrium (for extensive form games), with the major difference being that players (or the mediator) does not observe past recommendations (even if they do actions). Can the authors comment about whether their method would apply to this variant of CE?


**Limitations:**

-

**Strengths And Weaknesses:**

This paper is fairly clearly written and easy to follow. There are a fair number of minor spelling or grammatical errors, but these do not impede the reader.

The target audience is relatively niche and probably only of interest to those interested in constrained markov games. The technical contributions (a primal dual algorithm) are somewhat “standard” but non-trivial and interesting in their own right. Both Theorems 1.1 and 1.2 are unsurprising for those familiar with general-sum markov (or extensive-form) games, but is of interest to those who come from a single-player MDP background. The form of the primal update (11) is somewhat disappointing given that we essentially solve for a new unconstrained CE in each iteration, but perhaps that is unavoidable.

Note: I am unable to comment much on the technical soundness of the authors’ claims, especially since this is a paper with lengthy proofs.

---

> ### Author Response · Authors · 2022-07-31
> **Authors' reply to jkm3**
>
> Thank you very much for reviewing our manuscript and providing valuable feedback. Below is a response to the review comments. We have submitted a revised version with all revisions marked in **"red"**. Please let us know if further clarifications are needed.
>
> Q1: For constrained Markov game, does existence of Nash equilibrium (NE) imply the existence of correlated equilibrium (CE)? Are there some settings where CE exists but NE does not exist?
>
> A: Great questions. Regarding the first question, the existence of NE implies the existence of CE because every NE is necessarily a CE. Regarding the second question, note that it has been shown that NE exists for constrained Markov games under the standard Slater's condition [1]. Therefore, in the same setting, non-existence of NE would imply violation of the Slater's condition, which we think already makes the constrained Markov game less meaningful.
>
> [1] E. Altman and A. Shwartz. Constrained markov games: Nash equilibria. In Advances in dynamic games and applications, pages 213–221. 2000.
>
>
> Q2: We can maximize social welfare while maintaining incentive compatibility. Is there a cleaner way of selecting between various constrained CE?
>
> A: Good question. The original goal of constrained Markov game is to maximize $V_0^{(m)}(\pi)$ (i.e., social welfare) for each agent $m$ under the constraints $V_{j}^{(m)}(\pi) \ge c_j^{(m)}$ (i.e., incentive compatibility). However, CE does not necessarily achieve the maximum welfare under the constraints. Specifically, $\pi$ being a CE only means that no agent can benefit by modifying its policy alone under the constraints. Therefore, if we have multiple constrained CE policies, we can select among them based on their welfare values {$V _ 0 ^ {(m)}(\pi)$} $ _ {m=1} ^ M$. For example, we may select the CE that achieves the highest agents' average welfare $\overline{V}(\pi):=\frac{1}{M}\sum _ {m=1} ^ M V_0^{(m)}(\pi)$, or select the CE that achieves the lowest inter-agent welfare variance $\frac{1}{M}\sum _ {m=1} ^ M [V_0^{(m)}(\pi)-\overline{V}(\pi)]^2$, or select CE by considering both aspects.
>
>
> Q3: Does our method apply to CE of extensive-form games where agents have no access to past recommendations?
>
> A: Great question. We think our method can be extended to find CE of constrained extensive-form games. To elaborate, note that extensive-form games can be seen as a special case of partially-observable Markov game with a tree-structure assumption [2], and their CE is defined in a similar way. Also, there is an existing algorithm that finds CE of unconstrained extensive-form games with finite-time convergence guarantee [2]. Thus, a natural idea is to adapt our primal-dual algorithm by changing the subroutine algorithm (for finding CE of unconstrained Markov games) to the algorithm developed in [2]. We expect that the remaining proof does not rely on the structure of policy itself and thus follows similar proof logic to that of our Theorem 3.
>
> [2] Z. Song, S. Mei, and Y. Bai. Sample-Efficient Learning of Correlated Equilibria in Extensive-Form Games. ArXiv:2205.07223, 2022.

---

> > ### Comment · Reviewer_jkm3 · 2022-08-08
> > **Sorry for the late response**
> >
> > Apologies for the late response.
> >
> > 1. I see. This assumption on Slater's condition does feel a little strong for my tastes; ideally we should know if the constrains are infeasible. But its okay overall. I was more worried about the case where there *does* exist a constrained CE, but if becomes infeasible if one were to be restricted to independent strategies.
> >
> > 2. So, this means that to perform selection we have to keep training policies and pick the "best" policy from a finite number of them? That is disappointing, but I can understand thats not the main point of the paper.
> >
> > 3. Okay.
> >
> > I do not have too much more to add, unfortunately. I am generally happy with the paper, but due to my limited background I am unable to make a strong nor confident judgement. I am therefore keeping my score the same (a good thing).

---

> > > ### Author Response · Authors · 2022-08-08
> > > **Follow-up response to reviewer jkm3**
> > >
> > > Thank you very much for the follow-up response and appreciating our work.

---

### Official Review · Reviewer_zKK3 · 2022-07-11

**Rating:** 6
**Confidence:** 3
**Soundness:** 4 excellent
**Presentation:** 3 good
**Contribution:** 2 fair

**Summary:**

This paper deals with correlated equilibria in constrained Markov games over a finite horizon. The authors provide a sound definition of correlated equilibria for these type of games  that differs from previous definitions and show how to compute them using a primal-dual algorithm, similar to existing algorithms, albeit in a different context.

**Questions:**


The fact that the  stochastic and deterministic cases differ when constraints are added should not come as a surprise.
The same is true for Makov processes with one player: With no constraints, the optimal policy is deterministic while for Markov decision processes with linear constraints, the optimal policy are randomized. Actually, in this case, the number of randomizations is equal to the number of constraints. One wonders is a similar property holds here.





**Limitations:**

This work is mainly theoretical with no direct societal impact. However, as mentioned in the strength/weaknesses section, a motivation and a discussion on possible applications of this work is missing.

**Strengths And Weaknesses:**

Strengths:
- One contribution of this paper is a proper definition of correlated equilibria in Markov games over a finite horizon. The new definition is both natural and general.

- The idea of using measures as the state is nice and elegant: It transforms the problem into a linear one.

- The analysis of the performance of the algorithm is state of the art, even if the dependence in $H^9$ looks rather extreme, so that the performance of the algorithm in practice will be limited.




Weaknesses:

- The difference with previous definitions of correlated equilibria is mentioned several times but the reader fails to see exactly what  this difference is. The authors could have given the former definition  formally so that the reader can grasp this difference precisely. Here this is kept informal.
- This confusion lasts when the algorithm is presented as the first one to compute the correlated equilibria. If the definition is new and the notion of correlated equilibria is also different from previous attempts, then why is this algorithm qualified as the first one? It should be by essence. The same remark holds for Theorem 2 on strong duality. If the definition is new, surely strong duality has not been established before.

- The restriction to finite horizon games is definitely a limitation. Why have the discounted and/or long run behavior not been investigated? The finite horizon case is easier technically than both of these cases, and also less useful in practice.

- The motivation is lacking in the following sense: What kind of applications do the authors have in mind for constrained Markov games.
The brief mention of ressource constraints is not satisfying because these are usually hard constraints and not constraints that only hold in expectation as this is the case here. So far, this reviewer could not think of  any clear application to this framework.



-Finally, a discussion on the limitations of the last result on the sampling complexity  could be added. The sample complexity is only given for the expected gap and the expected contraint violation instead of for the true values. Also the bound is $H^9 \varepsilon^{-4} $ is not compared with similar cases.

---

> ### Author Response · Authors · 2022-07-31
> **Authors' reply to Q1-4 of zKK3**
>
> Thank you very much for reviewing our manuscript and providing valuable feedback. Below is a response to the review comments. We have submitted a revised version with all revisions marked in **"red"**. Please let us know if further clarifications are needed.
>
> Q1: Is the number of randomizations of a CE policy equal to the number of constraints for constrained Markov game?
>
> A: Good question. We conjecture that this may not hold for a CE policy of constrained Markov games, as the agents' actions are generally correlated. On the other hand, we think this may hold for a NE policy of constrained Markov games, as each agent's NE policy is independent from other agents' NE policies and hence can be viewed as the optimal policy of a single-agent constrained RL problem (by viewing all the other agents' NE policies as part of the environment). We will leave it to the future work to develop a rigorous proof.
>
> Q2: Should present the formal definition of the existing CE for constrained Markov game in [1], and show exactly the difference.
>
> A: Thanks for the suggestion. We have added more clarification on this point after Definition 3.2.
> To briefly explain the difference, our proposed CE in Definition 3.2 is defined for the original **constrained** Markov game, and requires both $\pi$ and the modified policy $\phi^{(m)}\circ\pi$ to be feasible. As a comparison, the existing CE proposed in [1] is defined for the **unconstrained** Markov game associated with the Lagrange function (i.e., with surrogate reward $R_{\lambda}^{(m)}(s_h,a_h):=r_0^{(m)}(s_h,a_h)+\lambda^{(m)\top} r^{(m)}(s_h,a_h)$), which does not necessarily require $\pi$ and $\phi^{(m)}\circ\pi$ to be feasible. Therefore, the existing CE proposed in [1] essentially follows the CE of **unconstrained** Markov games, whose definition is stated after our Definition 2.1. Moreover, [1] defined CE based on the state-action value function and consider stationary Markov policies in the discounted infinite horizon setting, whereas we define CE based on the state value function and consider non-Markov policies in the finite horizon setting.
>
> [1] V. Hakami and M. Dehghan. Learning stationary correlated equilibria in constrained general- sum stochastic games. IEEE Transactions on Cybernetics, 46(7):1640–1654, 2015.
>
>
> Q3: Why is this algorithm qualified as the first one that provably converges to CE of **constrained Markov games**? How about our strong duality result?
>
> A: As we clarified in the answer to Q2, the existing CE proposed in [1] is defined based on the **unconstrained** Markov game associated with the Lagrange function, while our CE is the first notion directly defined for **constrained** Markov games. In this sense, our algorithm is the first one that provably converges to this new CE. Similarly, our strong duality result is also the first one based on this new CE.
>
>
> Q4: Why have the discounted and/or long run behavior not been investigated?
>
> A: Good question. Our primal-dual algorithm can be generalized to handle the infinite horizon setting by changing the subroutine for solving the unconstrained Markov game. To explain, first, the strong duality result can be straightforwardly generalized to the infinite-horizon setting, as the structure of the linear program is highly similar to that in the finite horizon setting. Then, in each primal update, we will need to solve an infinite-horizon unconstrained Markov game similar to the one shown in eq.(8). To do this, we can extend the existing V-learning algorithm for finite-horizon Markov games to the infinite horizon setting. Specifically, the existing V-learning algorithm is based on the finite-horizon Bellman equation, and we can adapt it to the discounted infinite-horizon Bellman equation, leading to the update rule $V^{(m)}(s_t)\leftarrow (1-\alpha_t)V^{(m)}(s_t)+\alpha_t\big(r_t^{(m)}+\gamma V^{(m)}(s_{t+1})+\beta_t\big)$ (for agent $m$ at time $t$), where $\beta_t>0$ is the bonus to promote optimism and exploration.
> The convergence analysis of this primal-dual algorithm in the infinite-horizon setting follows the same logic as that of our current analysis. The only difference is that we will need to develop finite-time convergence analysis for the modified V-learning subroutine, which we think can be inspired from the existing analysis in the finite-horizon setting.

---

> ### Author Response · Authors · 2022-07-31
> **Authors' reply to Q5-7 of zKK3**
>
> Q5: What kind of applications for constrained Markov games? How can we formulate hard constrains into expectation form?
>
> A: Regarding the first question, for example, [2] formulates an anti-jamming wireless communication system as a constrained Markov game involving a legitimate transmitter and a jammer. Specifically, the state corresponds to how long has the transmitter get rid of the jammer. For the actions, the transmitter can select one of $K$ channels and a transmission rate to transmit message, and the jammer can select a channel and a power level to jam.
> Moreover, the jammer is subject to an average
> power constraint written as $\mathbb{E} _ {\pi}\big[\frac{1}{H}\sum _ {h=1} ^ {H}C_h(s_h,a_h^{(1)},a_h^{(2)})\big]\le P_{avg}$, where $C_h$ denotes the power consumption of jammer at time slot $h$ (we made their notations consistent with ours). This constraint can be rewritten into our standard form $\mathbb{E} _ {\pi}\big[\sum _ {h=1} ^ {H}r_{1,h}^{(2)}\big]\ge C_{\max}-P_{avg}$ by defining a reward for the jammer as $r_{1,h}^{(2)}=C_{\max}-C_h$, where $C_{\max}$ is an upper bound of $C_h$.
>
> Regarding the second question, if we have some additional hard constraints taking the deterministic form $r_{j,h}^{(m)}(s_h, a_h) \ge c_j^{(m)}$ for some $h, m, s_h, a_h$, our primal-dual algorithm can be extended to address them. Please refer to our response to the Q1 of Reviewer z16W for a comprehensive discussion. To summarize the main idea, the above hard constraints can be translated into a set of linear equality constraints $\pi_h^{(m)}(a_h|s_h)=0$ such that the violation $r_{j,h}^{(m)}(s_h,a_h)<c_j^{(m)}$ occurs for at least one $j$. In our response to Q1 of Reviewer z16W, we showed that these hard constraints can be rewritten into an expectation form by properly defining an auxiliary reward function. Moreover, we can adjust our primal-dual algorithm to handle these additional equality constraints by introducing additional dual variables.
>
> [2] M. K. Hanawal, M. J. Abdel-Rahman, and M. Krunz. Joint adaptation of frequency hopping and transmission rate for anti-jamming wireless systems. IEEE Transactions on Mobile Computing, 15(9):2247–2259, 2015.
>
> Q6: The sample complexity is only given for the expected gap and the expected constraint violation instead of for the true values.
>
> A: We want to clarify that the expectations in eqs.(14, 15) are only taken over the iteration $\widetilde{t}$, which is sampled from $\{0,1,\ldots,T-1\}$ uniformly at random. In another word, the expectation bounds in eqs.(14, 15) actually guarantee that
> $$\min_{0\le t\le T-1}D^{(m)}(\pi_t) \le \mathbb{E} \big[D^{(m)}(\pi_{\widetilde{t}})\big]:=\frac{1}{T}\sum_{t=0}^{T-1} D^{(m)}(\pi_t) \le 2\epsilon$$
> and that
> $$\min_{0\le t\le T-1}W^{(m)}(\pi_t) \le \mathbb{E} \big[W^{(m)}(\pi_{\widetilde{t}})\big]:=\frac{1}{T}\sum_{t=0}^{T-1} W^{(m)}(\pi_t) \le 2\epsilon.$$
> That is, there must exist output policy $\pi_t$ whose true values satisfy $\max\big(D^{(m)}(\pi_t),W^{(m)}(\pi_t)\big)\le 4\epsilon$.
> We hope this clarifies the reviewer's concern. Writing the bounds in such an expectation form has been widely adopted in the constrained RL literature [3,4].
>
> [3] D. Ding, K. Zhang, T. Basar, and M. R. Jovanovic. Natural policy gradient primal-dual method for constrained markov decision processes. In Proc. International Conference on Neural Information Processing Systems (Neurips), volume 2020, 2020.
>
> [4] Bai, Q., Bedi, A. S., and Aggarwal, V. Achieving Zero Constraint Violation for Constrained Reinforcement Learning via Conservative Natural Policy Gradient Primal-Dual Algorithm. ArXiv:2206.05850 (2022).
>
> Q7: The sample complexity bound is not compared with similar cases.
>
> A: Thank you for the great suggestion. We compare our sample complexity with that of two special cases of constrained Markov game. (1) For unconstrained Markov game, the state-of-the-art sample complexity $\widetilde{\mathcal{O}}(H^5 S A^{2} \epsilon^{-2})$ is achieved by V-learning [5], and the information theoretical lower bound is $\Omega(H^3SA\epsilon^{-2})$ [5]. (2) For single-agent constrained RL, the basic primal-dual algorithm that uses natural policy gradient in the primal update
> achieves the state-of-the-art sample complexity $\mathcal{O}(\epsilon^{-4})$ [4], which matches the dependence on $\epsilon$ of our sample complexity. We have added the above comparison at the end of Section 5.
>
> [5] C. Jin, Q. Liu, Y. Wang, and T. Yu. V-learning-–a simple, efficient, decentralized algorithm for multiagent RL. ArXiv:2110.14555, 2021.

---

### Official Review · Reviewer_z16W · 2022-07-11

**Rating:** 6
**Confidence:** 4
**Soundness:** 3 good
**Presentation:** 4 excellent
**Contribution:** 3 good

**Summary:**

The paper proposes a notion of correlated equilibrium (CE) for constrained Markov games, and a primal-dual algorithm to approximate one CE in such setting. The authors show that the algorithm guarantees duality gap and constraint violation of order $O(T^{-1/2})$. Finally, they study the sample complexity of the algorithm with a particular choice of subroutine for the primal update.

**Questions:**

[Q1] The motivating examples (e.g., lines 29-32) seem to be all related to settings with resource-consumption (i.e., packing) constraints. However, from eq. 3 and Assumption 2 (from which we have $r_{j,h}^{(m)}\ge 0$), my understanding is that the model only describes covering constraints (the cumulative r_{j,h} has to be at least a certain value). Is there any technical issue preventing you from addressing both kind of constraints? I don’t think that’s necessarily a problem, but it should be clarified.

[Q2] Regarding Assumption 1: assuming an exact knowledge of $\xi$ seems a bit restrictive. What would happen if the algorithm had access only to a lower-bound on the feasibility parameter $\xi$? Do you think the analysis would carry over? As a last comment regarding Assumption 1, I’m wondering what are its practical implications. In order to clarify this, it would be good to provide at least one practical example where such assumption is satisfied.

[Q3] On theorem 3. The authors claim that cumulative constraint violation is sublinear in T. However, I’m wondering what happens if the feasibility parameters $\xi$ of Slater’s condition are allowed to take arbitrarily small values. For example, what if $\xi_j^(m)$ were all equal to 1/T? It seems to me that the bound on constraint violation would not be sublinear in T. Is this correct? If so, I think that assumptions on the values allowed for $\xi$ should be better described.

[Q4] Is there any known lower bound to the sample complexity of the problem of Corollary 5.1? It not clear how tight is that bound.

**Limitations:**

limitations were discussed.

**Strengths And Weaknesses:**

The problem is interesting and the paper is clear and well written. The set of results provided in order to reach Theorem 3 is analogous to what is done in other works using primal-dual algorithms for online optimization with long-term constraints. However, I think the application of such techniques to the setting of constrained Markov games is novel and worth being studied.

There are a few points that should probably be clarified (see Questions). I’m willing to increase my score after reading the authors’ response (especially regarding Q3).

(one small typo at line 768: “it yield”)

---

> ### Author Response · Authors · 2022-07-31
> **Authors' reply to Q1, Q2 of z16W**
>
> Thank you very much for reviewing our manuscript and providing valuable feedback. Below is a response to the review comments. We have submitted a revised version with all revisions marked in **"red"**. Please let us know if further clarifications are needed.
>
> Q1: Is there any technical issue preventing you from addressing both packing constraints and covering constraints?
>
> A: Good question. Consider the hard packing constraints taking the form $r_{j,h}^{(m)}(s_h,a_h)\ge c_j^{(m)}$ for all $j,h,m,s_h,a_h$, which essentially constrain the selection of action $a_h$ in the state $s_h$. Therefore, the above hard packing constraints can be translated into a set of linear equality constraints $\pi_h^{(m)}(a_h|s_h)=0$ such that the violation  $r_{j,h}^{(m)}(s_h,a_h)<c_j^{(m)}$ occurs for at least one $j$. Note that the original covering constraints $V_{j}^{(m)}(\pi) \ge c_j^{(m)}$ are also imposed on policy $\pi$. Therefore, we just need to adjust our primal-dual algorithm to handle both the inequality and equality constraints on policy $\pi$. We elaborate it in the following aspects:
>
> (1) The Slater's condition becomes more stringent, i.e., the stochastic modification $\phi^{(m)}$ should satisfy not only the inequality $V^{(m)}(\phi^{(m)}\circ\pi)-c^{(m)}\ge \xi^{(m)}$ but also the above equality constraints;
>
> (2) Strong duality still holds. In the new Lagrange function, we add terms like $+\mu\pi_h^{(m)}(a_h|s_h)$ where $\mu\in\mathbb{R}$ is the Lagrange multiplier associated with the above equality constraint. Note that the equality constraint $\pi_h^{(m)}(a_h|s_h)=0$ is equivalent to the linear constraint on policy-induced joint measure $p_{\pi}(s_{1:h},a_{1:h})=0$, where the trajectory $\{s_{1:h},a_{1:h}\}$ contains at least one infeasible state-action pair $s_h,a_h$ (i.e., $r_{j,h}^{(m)}(s_h,a_h)<c_j^{(m)}$). Hence, the Lagrange function is still linear w.r.t. $p_{\pi}$, which a key property required for proving strong duality;
>
> (3) The algorithm is similar to the current version. For the primal update in eq. (11), note that the constraint $\pi_h^{(m)}(a_h|s_h)=0$ can be equivalently written as a covering constraint $\mathbb{E} _ {\pi} \big[\sum _ {h=1} ^ {H}r _ {eq,h} ^ {(m)}$ $(\widetilde{s} _ {h},\widetilde{a} _ {h})\big]=0$ where the equality-related reward $r _ {eq,h} ^ {(m)}(\widetilde{s} _ {h},\widetilde{a} _ {h})$ is defined as follows: $r _ {eq,h'} ^ {(m)} (\widetilde{s} _ {h'},\widetilde{a} _ {h'})=1$ if $h'=h$ and $(\widetilde{s} _ {h'},\widetilde{a} _ {h'})=(s_h,a_h)$; Otherwise, $r _ {eq,h'} ^ {(m)} (\widetilde{s} _ {h'},\widetilde{a} _ {h'})=0$.
> Therefore, we just need to add $\mu r _ {eq,h} ^ {(m)}(s_{h},a_{h})$ to the surrogate reward. For the projected dual descent step in eq. (12), the update for $\lambda_t^{(m)}$ remains the same whereas the update for $\mu\in\mathbb{R}$ uses gradient descent $\mu\leftarrow\mu-\eta\pi_h^{(m)}(a_h|s_h)$ without projection;
>
> (4) The remaining convergence proof follows the same logic as the current proof.
>
> Q2: What if we only know a lower bound of $\xi$? In what practical examples will Assumption 1 hold?
>
> A: Good questions. For the first question, suppose we only know a lower bound $\xi'$ that satisfies $0<\xi'\le \xi$, then all our results still hold by replacing $\xi$ with $\xi'$. To explain more specifically, note that the Slater's condition in Assumption 1 still holds with the lower bound $\xi'$.
> Therefore, the convergence rate in eq.(14) of the algorithm still holds. However, the constraint violation bound in eq.(15) will depend on $(\xi')^{-2}$ instead of $\xi^{-2}$, i.e., a loose estimate of the slackness will lead to a loose estimate of the the constraint violation convergence rate.
>
> For the second question, note that the Slater's condition holds if and only if $\inf_{\pi}\sup_{\phi^{(m)}}V_j^{(m)}(\phi^{(m)}\circ\pi) > c^{(m)}$ for all agents $m$. Intuitively speaking, whatever policies the other agents use, agent $m$ can always modify its own policy to make the joint policy feasible. In the work [1], they provided an example of anti-jamming wireless system that satisfies the Slater's condition. The system is formulated as a zero-sum constrained Markov game between a legitimate transmitter and a jammer. Specifically, the state corresponds to how long has the transmitter get rid of the jammer. For the actions, the transmitter can select one of $K$ channels and a transmission rate to transmit message, and the jammer can select a channel and a power level to jam. The jammer is subject to average power constraint that can be formulated as covering constraints. Under practical choices of transition kernel, rewards, etc., these constraints have been proved to satisfy the strong Slater's condition, as formally stated in Theorem 1 of [1].
>
> [1] Joint adaptation of frequency hopping and transmission rate for anti-jamming wireless systems. IEEE Transactions on Mobile Computing, 15(9):2247–2259, 2015.

---

> > ### Comment · Reviewer_z16W · 2022-08-08
> > **Follow up on Q1**
> >
> > Thanks for the detailed response.
> >
> > I'm still missing one point related to Q1 above. I understand how you can generalize from soft to hard constraints of the type $r(s,a)\ge c$. However, I was trying to understand whether you can handle the kind of constraints that would be needed in the examples that you mention:
> >
> > "users in a distributed power network have limited power budget [4], the users in a wireless communication network are assigned to limited bandwidth [28, 63]"
> >
> > In those cases it seems like you would need to define a resource-consumption function $f_t$ and have constraints of the form $f_t(s,a)\le c$, right? Can you manage constraints of that type in your framework? If not, I don't think that's a problem, but you should change the examples or explicitly state the kind of constraints you can handle within the framework.

---

> > > ### Comment · Reviewer_zKK3 · 2022-08-08
> > > **Agree**
> > >
> > > I agree with you, I still think that adding constraints only in expectaction seems too weak to cover hard ressource constraints.

---

> > > > ### Author Response · Authors · 2022-08-08
> > > > **Follow-up response to Reviewer zKK3**
> > > >
> > > > We would like to thank the reviewer for the follow-up comment.
> > > >
> > > > We agree with you and Reviewer z16W that the setting considered in this paper cannot directly handle hard constraints. In the new revision, we have replaced the examples mentioned in the previous introduction by some new examples that exactly considers soft (covering) constraints. We also further clarified the setting of the constraints in the revised Section 3.1.
> > > >
> > > > On the other hand, in our previous response to you and Reviewer z16W, we mentioned that our current primal-dual framework can be extended to handle hard constraints. The main idea is to show that any hard inequality constraint can be transformed into a soft equality constraint, in addition to the soft inequality constraints considered in this work.
> > > > We believe this is an interesting direction that deserves exploration in the future.

---

> > > ### Author Response · Authors · 2022-08-08
> > > **Follow-up response to Reviewer z16W**
> > >
> > > We would like to thank the reviewer for the follow-up question.
> > >
> > > Regarding the resource-consumption constraint mentioned by you, the resource-consumption constraint $f(s,a)<c$ can be rewritten as the hard constraint $r(s,a) > -c$ by defining the reward function $r(s,a) = -f(s,a)$ (reward can be understood as negative of cost). Therefore, the approach mentioned in our response to Q1 can handle this constraint, but requires extending our current framework to include covering equality constraints. We think this is an interesting direction that deserves exploration in the future.
> > >
> > >
> > > We also agree with the reviewer that the examples mentioned in our paper do not exact fit into our framework, and we have updated the paper with the following two revisions.
> > >
> > > **First**, we replace those examples by mentioning the following two examples that explicitly adopt the covering constraints considered in this paper.
> > >
> > > Example 1 (page 255 of [1]): In an uplink TDMA cognitive radio network where multiple cognitive radios (secondary users) attempt to access a spectrum hole, each user's total latency should not exceed a pre-specified threshold. This covering constraint is expressed as $ D^{(m)} (\pi) := \mathbb{E} _ {\pi} [\sum _ {h=1} ^ H d_h^{(m)}(s_h,a_h^{(m)})]\le \widetilde{D} _ i$ where $d_h(s_h,a_h^{(m)})$ denotes the $m$-th user's latency at time $h$ and $\widetilde{D}>0$ is the threshold. This constraint can be transformed to the constraint of our form $V^{(m)}(\pi):=\mathbb{E} _ {\pi} [\sum _ {h=1} ^H r_{h}^{(m)}(s_h,a_h)\le c^{(m)}]$ by letting $r_{h}^{(m)}(s_h,a_h):=-d_h^{(m)}(s_h,a_h^{(m)})$ and $c^{(m)}:=-\widetilde{D} _ i$.
> > >
> > > Example 2 ([2]):
> > > This work formulated an anti-jamming wireless system as a zero-sum constrained Markov game between a legitimate transmitter and a jammer. The jammer aims to jam the transmitter under average power constraint $C(\pi):=\mathbb{E} _ {\pi} [\sum _ {h=1} ^ H c_h^{(m)}(s_h,a_h)\le J_{\text{avg}}]$, where $d_h(s_h,a_h^{(m)})$ denotes the power level selected by the jammer at time $h$ and $J_{\text{avg}}>0$ is a threshold. This is also a covering constraint that has the same form as Example 1.
> > >
> > > [1] Y. Zhang and M. Guizani. Game theory for wireless communications and networking. CRC press, 2011.
> > >
> > > [2] M. K. Hanawal, M. J. Abdel-Rahman, and M. Krunz. Joint adaptation of frequency hopping and transmission rate for anti-jamming wireless systems. IEEE Transactions on Mobile Computing, 15(9):2247–2259, 2015.
> > >
> > > **Second**, when we introduce constrained Markov game in Section 3.1, we explicitly state that this paper considers covering constraints.
> > >
> > > Please let us know if you are satisfied with these updates or you have other questions.

---

> > > > ### Comment · Reviewer_z16W · 2022-08-09
> > > > **One last comment**
> > > >
> > > > Thanks for the response. I think the paper is clearer now that it clearly states what constraints it can handle. I will update my score to a 6.
> > > >
> > > > One last comment: I think one obstacle to the approach which you propose to handle resource consumption constraints is that Assumption 2 prevents you from setting $r(s,a)=-f(s,a)$.

---

> ### Author Response · Authors · 2022-07-31
> **Authors' reply to Q3, Q4 and typo of z16W**
>
> Thank you very much for reviewing our manuscript and providing valuable feedback. Below is a response to the review comments. We have submitted a revised version with all revisions marked in **"red"**. Please let us know if further clarifications are needed.
>
> Q3: What if $\xi$ in the slackness condition is arbitrarily small (e.g., scale as $1/T$)?
>
> A: We want to clarify that the slackness $\xi$ is a fundamental parameter of the underlying constrained Markov game and is independent of the algorithm. Hence, $\xi$ will not depend on the number of iterations $T$ of the algorithm.
> More specifically, based on Assumption 1, the slackness of agent $m$'s $j$-th constraint is defined as
> $\xi_j^{(m)}:=\inf_{\pi}\sup_{\phi^{(m)}}V_j^{(m)}(\phi^{(m)}\circ\pi)-c^{(m)}$, which is a constant that only depends on the constrained Markov game setting itself and does not depend on any parameter of the algorithm (such as $T$). We think the reviewer may mistake the $\inf_{\pi}$ part as a specific policy $\pi_t$ at iteration $t$, and hope this clarifies the reviewer's concern.
>
> Q4: Is there any known lower bound to the sample complexity of the problem of Corollary 5.1?
>
> A: Good question. In the existing literature, an information theoretical lower bound $\Omega(H^3SA\epsilon^{-2})$ for unconstrained Markov game (special case of constrained Markov game) has been established in [2]. We have added comparison of our complexity with this lower bound after Corollary 5.1.
> To the best of our knowledge, no lower bound has been established for constrained Markov games, for which we think the complexity lower bound is in general higher.
> We believe this is an interesting future work, and the complexity lower bound may critically depend on the structure of the constraints.
>
> [2] C. Jin, Q. Liu, Y. Wang, and T. Yu. V-learning–a simple, efficient, decentralized algorithm for multiagent RL. ArXiv:2110.14555, 2021.
>
> Q5: Small typo at line 768: "it yield".
>
> A: Thank you for pointing this out. We have corrected the typo and rewritten that sentence.

---

### Official Review · Reviewer_ybg9 · 2022-07-12

**Rating:** 6
**Confidence:** 2
**Soundness:** 3 good
**Presentation:** 3 good
**Contribution:** 3 good

**Summary:**

This paper studies correlated equilibrium in Markov games with linear constraints. The authors start with a strong duality theorem for CE in constrained Markov games, which enables them to design a primal-dual algorithm that uses a unconstrained CE algorithm as a subroutine. The primal-dual algorithm is shown to have a $O(1/\sqrt{T})$ convergence rate in $T$ iterations, and when V-learning is used as the solver, an $O(H^9SA^2\epsilon^{-4})$ sample complexity bound.

**Questions:**

Is it possible to output an approximate CE that is strictly feasible? The answer seems to be true for constrained MDPs.

**Limitations:**

Limitations and potential societal impact have been adequately addressed.

**Strengths And Weaknesses:**

Strengths:
- The paper establishes a reasonable formulation for correlated equilibrium in constrained Markov games, which captures this important concept and as far as I know is novel.
- The proposed primal-dual algorithm is compatible with essentially any black-box CE solver.
- The theoretical results seem sound and novel.

Weaknesses:
- The $O(1/\epsilon^4)$ sample complexity bound seems a bit loose, given that works in constrained MDP provide $O(1/\epsilon^2)$ rates. It might be due to V-learning being used as a black-box algorithm that is initialized tabula rasa each step. It seems to me that this can be improved by using a less black-box approach, perhaps by using model-based algorithms.

---

> ### Author Response · Authors · 2022-07-31
> **Authors' reply to ybg9**
>
> Thank you very much for reviewing our manuscript and providing valuable feedback. Below is a response to the review questions/comments. We have submitted a revised version with all revisions marked in **"red"**. Please let us know if further clarifications are needed.
>
> Q1: Is it possible to output an approximate CE that is strictly feasible?
>
> A: Good question. We think it is possible to leverage the conservative constrained RL framework proposed in [1] to
> modify our algorithm and outputs a strictly feasible approximate CE.
>
> To elaborate the main idea, instead of considering the constraints $V_j^{(m)}(\pi) \ge c_j^{(m)}$ of the original constrained Markov game in eq.(3), we can consider a more conservative game that adopts stronger constraints, i.e., $V_j^{(m)} (\pi) \ge c_j^{(m)} + \delta$, where $\delta>0$ is a tuning parameter. Note that introducing such a parameter $\delta$ does not change the overall structure of the constrained Markov game, and hence the strong duality result still hold as long as $\delta<\min_{j,m}\xi_j^{(m)}$. In the single-agent case, [1] developed a primal-dual type
> algorithm for solving the conservative constrained RL problem with a proper choice of $\delta$, and showed that the output policy is a strictly feasible policy for the original single-agent constrained RL problem. We think it is possible to generalize this approach to constrained Markov games as strong duality is preserved, and we leave the technical developments for future work.
>
> [1] ``Achieving Zero Constraint Violation for Constrained Reinforcement Learning via Primal-Dual Approach''. AAAI 2022.

---

### Meta-Review · Area_Chair_P39J · 2022-08-25

**Recommendation:** Accept
**Confidence:** Less certain

**Metareview:**

This paper proposes and examines a notion of correlated equilibrium for "constrained stochastic games", that is, stochastic games where the players seek to optimize their payoffs modulo guaranteeing a certain target.

The reviewers' initial concerns were addressed satisfactorily by the authors during the rebuttal phase, leading to a unanimous "accept" recommendation from the reviewers. After my own reading of the paper, I concur with this assessment: the paper treats an interesting and timely topic, and the results are both interesting and technically challenging. On a personal note, I would urge the authors to explain in more detail the notion of a "constrained" Markov game, as the terminology is not quite standard in game theory (where constraints typically have a different meaning than in MDPs); however, other than that, the reviews speak for themselves and I am also happy to recommend acceptance.

**Award:**

No

---

### Decision · Program_Chairs · 2022-09-14

Accept